# The temporal variation in pesticide concentrations within matured French wines

**Pieter Spanoghe**[1]*, **Jasmine De Rop**[1], **Lilian Goeteyn**[1], **Agnieszka Deja-Muylle**[1], **Hannah Vanderstappen**[1], **Lotte Neckebroeck**[1], **Dries Verhaegen**[1], **Pauline van den Hove**[1], **Joachim Neri**[2], **Erik Meers**[2]

1 Department of Plants and Crops, Crop Protection Chemistry, Gent, Belgium, 2 Department of Green Chemistry & Technology, Resource Recovery in the Biobased Economy, Gent, Belgium

* Pieter.Spanoghe@UGent.be

**Data Availability Statement:** All relevant data are within the manuscript and its Supporting Information files.

## Abstract

Numerous organizations worldwide are diligently working to regulate the composition of food products, with a particular focus on pesticide content. Each year, several substances are classified as hazardous to human health and subsequently banned from agricultural use. In this study, we address the age-old question: "Does wine improve with time?" from the context of pesticide composition. We gathered wine samples from renowned French winemaking regions, covering the years 1935 to 2000, to assess pesticide levels and identify specific substances. Our objective was to determine if any currently banned pesticides were present in these aged wines and whether the detected levels pose health risks under typical daily consumption patterns. Our findings revealed the presence of trace amounts of 21 different pesticides proceeding from Plant Protection Products (PPPs), in most of the wine samples, albeit at levels considered non-threatening to human health. Notably, one sample exhibited an alarmingly high concentration of carbaryl, surpassing toxic consumption thresholds. This study prompts discussions regarding the prioritization of pesticide testing in various products and whether stringent regulations should be upheld in the wine selling collectors sector.

## 1 Introduction

The vine (*Vitis vinifera*) is an economically significant crop [1] renowned for its grape production. Grapes serve various purposes, not limited to fresh consumption or drying, but also encompass the vital art of winemaking. It is widely acknowledged that agriculture, including wine cultivation, employs a range of plant protection products (PPPs) such as insecticides, fungicides, and herbicides [2]. These are integral to safeguarding cultivated crops and ensuring bountiful harvests.

Over the years, both in Europe and globally, there has been a growing awareness of the health and environmental implications associated with the extensive use of PPPs [3,4]. Consequently, new legislation has emerged, leading to increased restrictions on their usage (e.g. EU legislation regarding the bringing on the market of new PPP)https://eur-lex.europa.eu/legal-content/EN/TXT/?uri=CELEX%3A32009R1107. This shift in policy is driven by concerns

**Funding:** The author(s) received no specific funding for this work.

**Competing interests:** The authors have declared that no competing interests exist.

related to potential long-term effects on human health and the environment, as well as the enduring presence of some persistent PPP residues in our surroundings long after their application [5]. The aim is to strike a balance between agricultural productivity and the well-being of ecosystems and human populations.

Before 1991, there was a lack of uniformity in legislation across European Union (EU) Member States regarding the use of PPPs. However, in 1991, the European Commission initiated a pivotal shift by introducing the first directive (Directive 91/414/CEE) aimed at regulating the authorization of pesticides for use in European agriculture. In 2009, this legislation underwent a comprehensive revision, leading to the current regulation (AG) No 1107/2009, which remains in effect today [6,7]. This updated framework places a strong emphasis on the safe application of PPPs, ensuring their compatibility with all life forms on Earth, including humans and animals, while safeguarding the environment. It also introduces guidelines for the sustainable utilization of these products. Guidance sets what maximum residues of PPPs can be present in agricultural products. Each agricultural product is tested for the presence of PPP before consumption and that amount (residue) is compared with Europe's MRL (Maximum Residue Level) database. Approved pesticides undergo periodic re-evaluation and may be either retained or banned from agricultural use, resulting in a dynamic landscape of authorized PPPs. Pesticides usage, safety, influence on environment and finally their content in food products are also important topics of The European Green Deal that sets crucial goals for climate till year 2050 [8].

Awareness of food safety, especially in alcoholic beverages, has increased significantly in recent years. Pesticide use in wine production is a major concern due to its ecological and health implications. Analytical methods for assessing pesticide levels primarily target grapes, with techniques such as high-performance liquid chromatography (HPLC) and gas and liquid chromatography (GC, LC) being commonly employed. Sample preparation methods vary and include QuEChERS, solid-phase microextraction (SPME), pressurized liquid extraction (PLE), dispersive liquid-liquid microextraction (dLLME), and accelerated dispersive liquid-liquid microextraction (ADLL-ME) [9]. These methods are also applied to analyze the final consumer product, including both the liquid and sediment phases of wine. For instance, [10] identified 48 pesticides in wine samples using liquid chromatography-tandem mass spectrometry (LC-MS/MS). [11] introduced a green solvent-based DLLME/HPLC-MS method for extracting and analyzing pesticide content. More recently, [12] validated QuEChERS extraction combined with Gas Chromatography-Mass Spectrometry (GC-MS) as the most optimized method for testing pesticide content in wine, applying their approach to both grape and wine phases. Research in this area extends beyond commercially available wines to include organic varieties, as evidenced by studies such as [13]. This comprehensive approach ensures that pesticide levels are accurately monitored throughout the production process, from grape cultivation to the final product.

Maximum Residue Limits (MRLs) for pesticide active ingredients are established globally to ensure the safety of food consumption. According to a 2016 European Union report on pesticide residues, the incidence of MRL exceedance in wine is relatively low [14]. However, wine production begins at the vineyard, where the frequency of pesticide residues in untreated grapes is typically high. It's crucial to consider that processing factors also affect pesticide levels. Processed grapes, such as those used in making raisins, often show some of the highest frequencies of MRL exceedances among processed foods [15]. Therefore, when evaluating MRLs in wine, it's essential to account for processing factors. These factors not only reflect the impact of processing on the food product but also on the chemicals themselves. This underscores the importance of continuous monitoring and evaluation throughout the wine production process, from the grapes in the vineyard to the final bottled product. However, stored products

evade further evaluation over time. This can result in a lack of control over chemicals that accumulate in wine, highlighting a significant gap in ensuring ongoing safety and compliance.

For consumers, savoring a glass of wine does not typically conjure thoughts of the pesticides used in vineyards. Although studies have been made to indicate the pesticide content in wines available currently on the market [13], trend in consumption is more focused on organic values than possible, health influencing, pesticide content [16]. Similarly, consumers may not consider the possibility of residual traces of older PPPs, which are no longer authorized for use on grapes, persisting in the wine over time. This research endeavors to investigate the presence of such traces in 84 French wines produced before the year 2000. This study not only examines the wine, which consumers drink, but also delves into the solid phase, including wine sediment and lees, left behind in the bottle. In addition to detection of 21 pesticides, the study assesses the presence of copper residues, a significant inorganic component renowned for its fungicidal properties and use in viticulture. Furthermore the risk assessment of detected levels of pesticides was assessed for different age groups and adults versus children. This allowed insight into toxicity due to causal and prolonged usage.

## 2 Material and methods

### 2.1. Sampling and clean-up extraction of pesticides from wine and wine sediment

A total of 84 bottles of French red wines were acquired from wine collectors within the author's personal network, including family and friends. This study primarily focused on older wine production, spanning from the years 1937 to 2000, thereby covering a significant historical range. The sample set was thoughtfully selected to represent various wine-growing regions in France, and a detailed breakdown can be found in the (S1 Table). Fig 1 illustrates the distribution of wine bottles by their age.

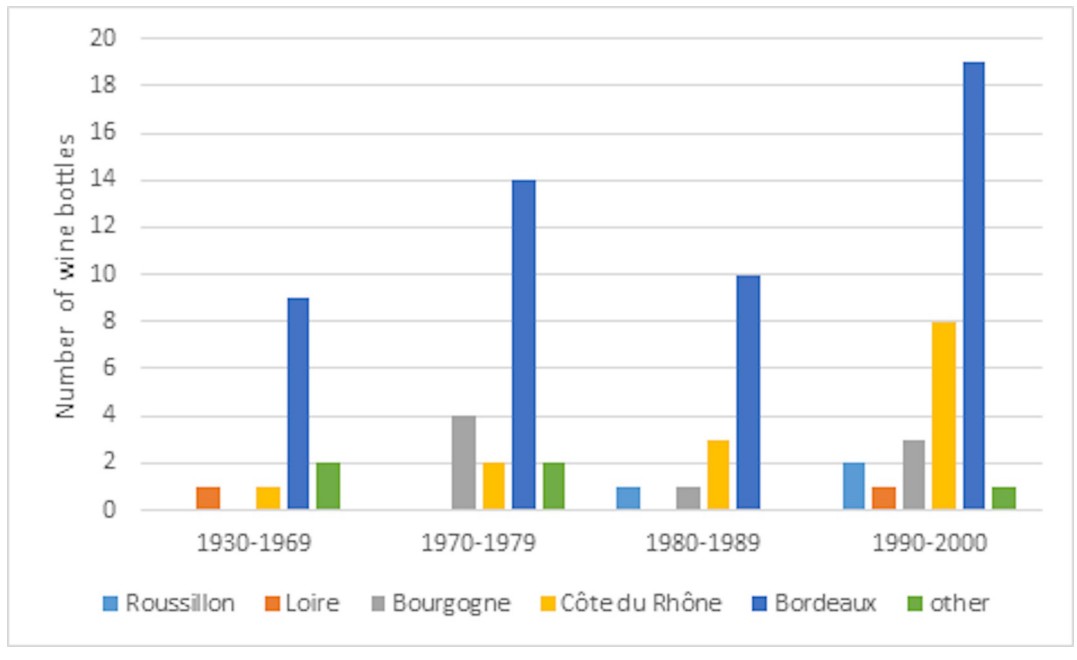

**Fig 1. The distribution of wine bottles by year of production and region of production in France.**

Each of the wine samples underwent testing for the presence of pesticides—residues stemming from PPPs. Each bottle provided two samples: one from wine and the other from the solid phase, comprising wine sediment and lees. The bottle contents were filtered to separate the wine from the solid components, including lees. This filtration process involved passing 400 to 500 mL of wine through filter paper (Whatman GF/D glass microfiber filters, 47mm diameter), leaving behind any residual matter.

The filtered wine liquid phase underwent solid-phase extraction, employing an efficient and robust technique based on the method previously described by [17]. Sep-Pak® C18 Classic Cartridges (Whatman Glass Microfiber Binder Free Filter, 2.7 Micron, 2.2 GF/D, 4.7cm Diameter) were employed in this process. The C18 cartridge was first activated with 2 mL of methanol HPLC grade (Chem-lab NV, 2.5L) and rinsed with 2 mL of HPLC water (Chem-lab NV, 2.5L). Subsequently, 50 mL of the filtered wine sample was diluted with 50 mL of Milli-Q water (in house dispenser) and passed through the cartridge. Following this step, 2 mL of HPLC grade hexane (Alltech Associates Inc, 2.5L) and 8 mL of HPLC grade acetonitrile (Chem-lab NV, 2.5L), were used as an eluent. This sample preparation method was adapted from the methodology outlined by [17]. It's worth noting that the dilution of wine samples was found to significantly impact extraction efficiency, with a 1:1 dilution ratio yielding superior results compared to undiluted wine samples. This effect is likely due to the high ethanol content typically found in wine. The residual material on the filter paper was allowed to air-dry to determine accurate lees masses.

The QuEChERS method, which stands for 'Quick', 'Easy', 'Cheap', 'Effective', 'Rugged', and 'Safe', was employed to extract PPPs from the wine sediment. This method consists of two main steps: first, the extraction of desired analytes using acetonitrile, followed by the purification of the extract through Solid-Phase Extraction (SPE), as outlined by [18].

The dried wine sediment, along with the filter paper, was transferred to a Falcon (Nerbe Plus, PP, 50 ml) tube and 15 ml of Acetonitrille was added. The tubes contained: 1.5 g of sodium citrate (Sigma-Aldrich, 1kg), 1.5 g of sodium chloride (Sigma-Aldrich, 1kg), 0.75 g of sodium hydride (Sigma-Aldrich, 1kg), and 6 g of magnesium sulfate (Sigma-Aldrich, 1kg). The salts are used as a buffer to maintain optimal pH during extraction in acetonitrile, while magnesium sulfate is used to remove water. The sample was vigorously shaken (manually) and centrifuged at 10,000 rpm for 5 minutes. Subsequently, 10 mL of the upper layer was pipetted off to a pre-prepared tube (DisQuE® 1200 mg, WATERS) containing $MgSO_4$, PSA (Primair Secundair Amine), and C18 were utilized. After thorough shaking, the PSA tubes were centrifuged at 10,000 rpm for 5 minutes. Subsequently, 5 mL of the purified sample was transferred to a flask for solvent exchange [18].

To facilitate LC-MS/MS analysis, the extracted samples from both the wine sediment and the wine were evaporated (Rotavapor) to exchange solvents. The PPPs were redissolved in 2 mL of a solution comprising 90% Milli-Q $H_2O$ and 10% ACN (acetonitrile) for LC-MS/MS analysis. The flask was briefly placed in an ultrasonic bath (Elma® Transsonic T700) before x ml was transferred to a vial.Finally, vials were filled for temporary storage at C degress, based on the analysis timeline.

A recovery correction factor was applied to the results to account for the matrix effects, which were significant in this case. Despite these effects, the method demonstrated adequate reproducibility. To evaluate the matrix, organic wine and tested on carbendazim, providing a representative sample for the analytical performance. A multi-residue method was employed to address the challenge that one method cannot extract all pesticides, with different polarity/solubility, in a single solvent, at 100% efficiency. This approach ensures comprehensive detection and quantification of various pesticides, compensating for the limitations inherent in solvent-based extraction methods.

To determine the presence and quality of copper content in wine samples, the wine matrix was subjected to a destructive process as per the procedure specified by Vito: WAC/III/B/001. This involved adding 4 mL of HNO3 to 30 mL of wine in a 100 mL beaker, which was then covered with a watch glass. The destruction process occurred over 2 hours at 120˚C and was quantitatively transferred to a 50 mL volumetric flask as outlined by [19].

## 2.2. LC-MS/MS analysis

Following sample collection and extraction, the wines were subjected to liquid chromatography analysis coupled with a triple quadrupole double mass spectrometry, using Model X LC-MS/MS and double mass spectrometry [20 For the analysis of organic Plant Protection Products (PPPs), LC-MS/MS method was used (model Xevo TQD Waters), utilizing an Acquity UPLC BEH C18 1.7μm column (2.1*100mm, Waters). Total of 98 pesticides was targeted with this method (S12 Table). The column temperature was maintained at 40˚C, while the desolvation temperature reached 450˚C. The mobile phase consisted of acetonitrile (solvent B) and water with 0.1% formic acid (Merck) (solvent C), flowing at a rate of 0.4 ml/min. Solvent B and C were distributed in varying proportions between 1% and 99% over time. Each run had a duration of 11 minutes.

It's worth noting that during the analysis, some pesticides from the wine samples may not have been completely extracted or could have been lost. Interactions between pesticides and other extracted components might have led to a matrix effect, as documented by [21]. To account for these variations and their impact on the detected concentrations, we determined the recoveries of the PPPs. The recovery values for the active ingredients in both wine and wine sediment, along with validation parameters like Limit of Detection (LOD) and Limit of Quantification (LOQ), can be consulted in S2 and S3 Tables.

The Limits of Detection (LOD) and Limits of Quantification (LOQ) were determined by injecting the lowest concentration eight times and by analyzing the signal-to-noise (S/N) ratio. The variability was assessed by repeating the measurements three times for the LOD and ten times for the LOQ. The quantification has been according to the standard curve. The fixed detection limits used are based on the lowest standard concentration injected. Quantification was performed based on pre and post spiking of the wine (pre-spiking of the wine, post–spiking of the wine extract). The residues were quantified with solvent standards. Only for carbendazim a matrix-matched standard was used for quantification. Range of calibration standard was: 0.0001 to 0,1 mg/L extract.

## 2.3. Copper content determination

To determine copper content in the wine samples, we utilized ICP-OES (Inductively Coupled Plasma Atomic Emission Spectroscopy), employing a Varian Vista MPX + autosampler in accordance with Vito: WAC/III/B/010 [22].

## 2.4. Food safety assesment

To determine whether a food safety issue is present when drinking aged wines, a risk analysis was performed. This involved a comparison of a daily intake of wine with the acceptable daily intake (ADI), of each substance, which represents the maximum daily intake of a substance that a person may consume over a long period without incurring health risks. The ADI is expressed in mg per kg body weight per day (mg/(kg BW*d)) and can be found for each pesticide in S4 Table. It refers to chronic toxicity or long-term exposure [23].

In subsequent calculations, only the liquid part of the wine was considered, with the assumption that consumers do not ingest the sediment of the wine. According to the latest

data from the European Food Safety Agency, the average wine consumption in Belgium is reported to be 300g/day, while frequent wine consumers reach 653g/day [24]. Calculations here presented are based on the assumption that the mass density of wine is equivalent to that of water, i.e., 1 kg/m$^3$. Belgian men are 1.76 m tall on average and weigh 79 kg. The average height for Belgian women is 1.64 m and the average weight is about 66.7 kg [25]. With these data, consumption per kg body weight was calculated according to the formula below.

$$Y\left(\frac{mg}{kg\ BW*d}\right) = X\frac{mg}{L\ wine}*a\ \frac{L\ wine}{day}*\frac{1}{b\ kg\ BW}$$

$$Y = Intake\ of\ wine\ per\ day\ and\ mean\ body\ weight\left(\frac{mg}{kg\ BW*day}\right)$$

$$X = mean\ concentration\ of\ positive\ sample > LOQ\left(\frac{mg}{L\ wine}\right)$$

$$a = average\ consumption\ of\ 300g|frequent\ consumption\ of\ 653\ g\ wine\ per\ day$$

$$b = ♀\ 66.7\ or\ ♂\ 79\ kg\ BW$$

## 3 Results and discussion

### 3.1. PPP residues in the wine liquid phase

**3.1.1. Types of PPPs detected.**   Among the 21 pesticides detected, 20 are quantifiable, as depicted in Fig 2. Carbendazim emerges as the most prevalent pesticide, being present in 88% of the samples, corresponding to 74 wine samples. Following closely is cadusafos, detected in 60% of the samples (50 wine samples). It is important to note that, unlike carbendazim, all cadusafos measurements were non-quantifiable, as the levels were below the limit of quantification (LOQ). Piperonylbutoxide and metalaxyl were also notable findings, appearing in 43% and 40% of the samples, respectively, amounting to 36 and 34 wine samples, respectively. For detailed information on the detected pesticides, including their mean concentrations above the LOQ, please refer to provided in the S5 Table.

According to [26], several of the pesticides detected in our study, namely azoxystrobin, difenoconazole, prochloraz, and tebuconazole, are primarily employed for the control of powdery mildew in Belgium. Azoxystrobin and difenoconazole also find application in addressing downy mildew, with additional support from metalaxyl, dimetomorph, and copper. Furthermore, these two substances are utilized in the control of grey mold. Pyrimethanil, tebuconazole, and diethofencarb can also be harnessed to combat this particular mold. When it comes to battling black rot, azoxystrobin, difenoconazole, tebuconazole, and metalaxyl are the go-to choices. Examining Fig 2, it becomes evident that a significant portion of the identified Plant Protection Products (PPPs) can be attributed to their necessity in safeguarding crops against downy mildew, powdery mildew, black rot, and grey mold. However, the presence of certain substances, such as carbendazim, cadusafos, and carbaryl, in many samples raises questions since these substances are no longer approved for use on grapes by the European Commission, aligning with current legislation. Notably, carbendazim, according to the study by [27], exhibited remarkable efficacy in reducing powdery mildew incidence by 75% in mangoes. It also displayed effectiveness in combating rot and downy mildew (APVMA, 2007). Despite its wide-ranging antifungal properties, carbendazim is no longer authorized due to its adverse

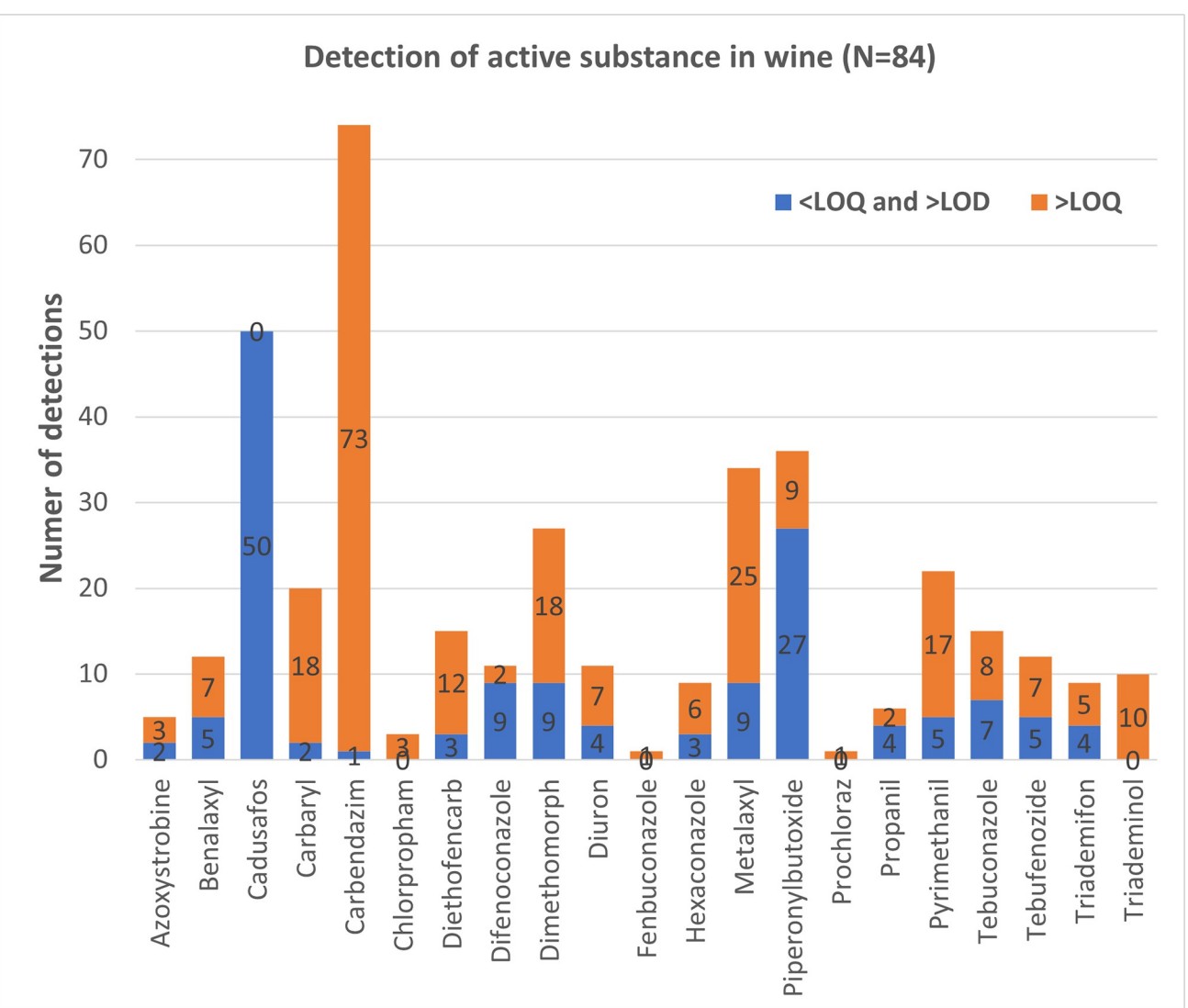

**Fig 2. The frequency of detection for each pesticide in the analyzed wine bottles, categorized by levels of detection.** Quantifiable concentrations (above the limit of quantification, >LOQ) and detectable concentrations (between the limit of detection, >LOD, and the limit of quantification, <LOQ).

impact on human health [28]. Given its broad-spectrum activity, it is reasonable to assume that this substance was frequently applied to protect *V. vinifera*. In contrast, cadusafos and carbaryl serve as insecticides rather than fungicides. They target nematodes, aphids, and butterflies, respectively. Furthermore, piperonyl butoxide, detected frequently, acts as a synergist for (organic) insecticides, indirectly contributing to the reduction of pest insect populations [29].

Plant protection products are categorized into several groups, with the primary ones being fungicides, insecticides, and herbicides. In the scope of this study, the majority of the identified pesticides exhibit fungicidal activity, totaling 14 out of the 21 pesticides. Additionally, we detected five types of insecticides, three types of herbicides, and one synergist that enhances the effectiveness of insecticides (please refer to S5 Table). Fig 3 provides a breakdown of the identified pesticides based on their respective types. Fungicides emerge as the predominant category, with 245 detections recorded across the 84 wine samples. In contrast, the second-highest number of detections pertains to insecticides, accounting for 82 observations. Notably,

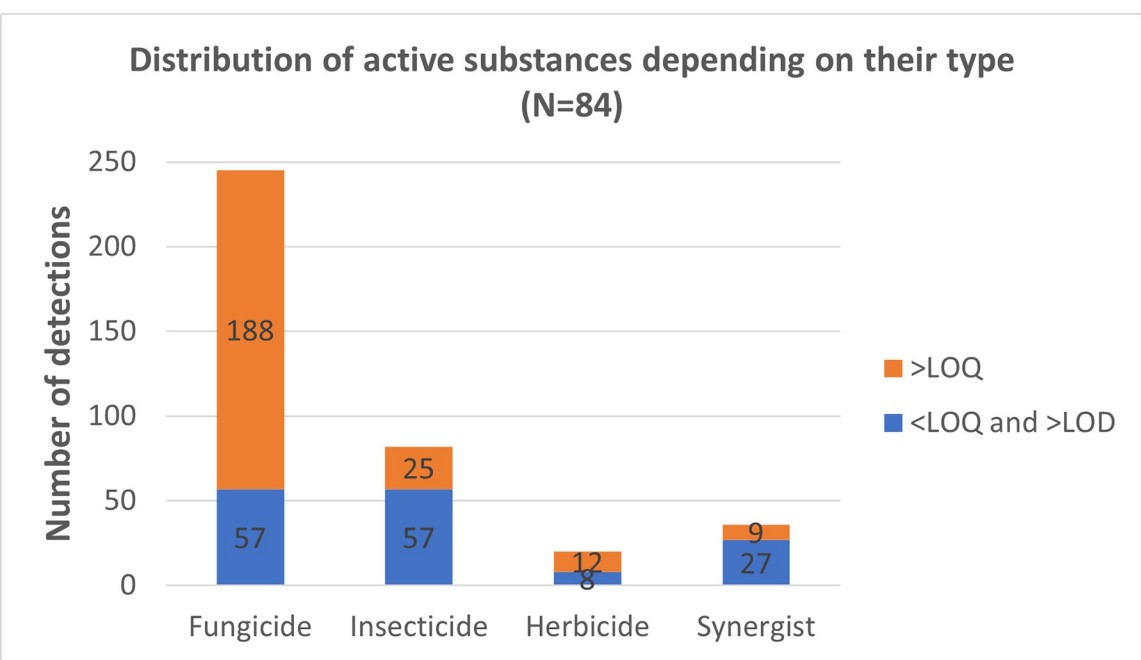

**Fig 3. The distribution of pesticides in the analyzed wine bottles, categorized by substance type.** Quantifiable concentrations (above the limit of quantification, >LOQ) and detectable concentrations (between the limit of detection, >LOD, and the limit of quantification, <LOQ).

the color distribution of the bars in Fig 3 reveals an interesting trend. Fungicides tend to occur primarily in quantifiable concentrations, constituting 77% of the total detections. Conversely, insecticides are more frequently found in lower concentrations, falling either below the LOQ or above the LOD, making up 63% of the detections in this range.

The prevalence of fungicides compared to other types of Plant Protection Products (PPPs) can be logically explained. Fungicides exert their effects systemically, targeting fungi present on or within the grapevine and its fruit. Moreover, *Vitis vinifera*, the grapevine species in focus, is notably more susceptible to fungal diseases compared to insect pests. In essence, fungicides are indispensable for managing fungal threats, as they act throughout the grapevine's entire structure, safeguarding both the plant and its fruit. In contrast, insecticides primarily address insect pests that often reside on the plant. Consequently, their action is primarily directed at the surface of the grapes, as they aim to eliminate insects without extensive systemic activity within the plant. Herbicides, on the other hand, are relatively less common in our findings. This can be attributed to their mode of action, which involves targeting and being applied to the plants situated between the vines rather than directly impacting the grapes themselves. The herbicides detected in quantifiable quantities (comprising 60% of the total herbicide detections) typically exhibit systemic properties, as they are absorbed through the root system of the vines and subsequently transported into the grapes.

Pesticides can be categorized into systemic and contact pesticides. Systemic pesticides are absorbed by the plant and operate by transport and redistribution, while contact pesticides act at the specific site where they have been applied. Fig 4 illustrates the total number of detections of systemic and contact PPPs across the 84 analyses. It is evident that systemic PPPs are more frequently detected than contact PPPs, in line with our expectations. Specifically, systemic PPPs were detected a total of 258 times, while contact PPPs were found only 89 times. Among the systemic PPPs, 76% of the total number of detected observations were quantifiable

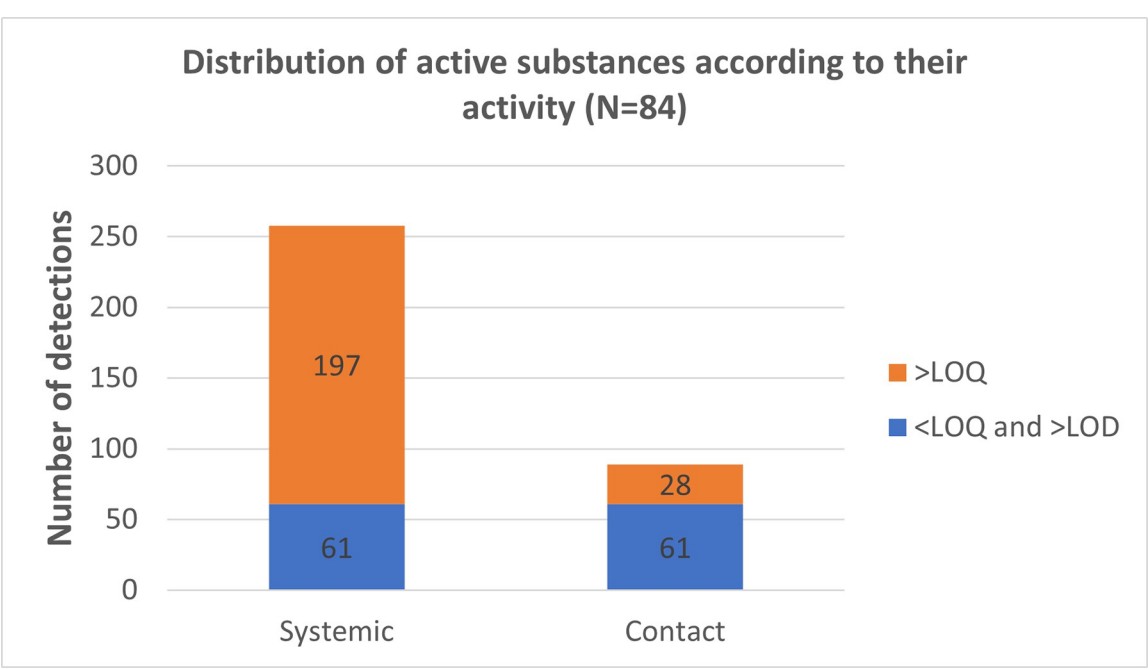

**Fig 4. The distribution of pesticides in the analyzed wine bottles, according to the substance type.** Quantifiable concentrations (above the limit of quantification, >LOQ) and detectable concentrations (between the limit of detection, >LOD, and the limit of quantification, <LOQ).

(>LOQ), whereas for the contact PPPs, only 31% were present at concentrations above the LOQ. This demonstrates that systemic PPPs are generally present in relatively higher concentrations compared to contact agents. This difference can be attributed to the fact that systemic PPPs penetrate the plant's cuticle and migrate into the plant tissue (such as the grape pulp), whereas contact PPPs are less prevalent in wines due to their susceptibility to being washed off by rain or runoff from leaves and grapes. Consequently, contact PPPs are often no longer present at the time of harvest, whereas systemic PPPs persist for a longer duration. It is worth noting that when wines mature in typical oak barrels, there is a possibility that PPPs from the wood of the barrels may permeate the wine, and vice versa. Consequently, an exchange of pesticides may occur between different wines stored in previously used barrels. Additionally, application rates and dosage might also very due to type of pesticides. For example, fields are usually more treated with fungicides than insecticides and sometimes even at the higher dosage.

**3.1.2. Number and amounts of PPPs detected.** The results of residue analysis in wine are presented in S6 Table. Out of the 84 wine samples analyzed, traces of 23 different pesticides were detected, with concentrations exceeding the LOD in at least one sample. 19 residues were detected in total in wine extracts. Except for seven wine samples, the majority of wine samples had at least one pesticide detected (>LOD). For residues above LOD, a distinction was made between detectable and quantifiable concentrations.

Across all wines, an average of 1.8 pesticides were found in detectable concentrations (but below LOQ), while 2.8 quantifiable substances were identified (as shown in S8 Table). On average, a total of 4.6 pesticides were identified per bottle. Fig 5 provides a comprehensive representation of the number of pesticides determined per bottle.

In 4 bottles (5%), exactly one pesticide was detected. In 57 bottles (68%), between 2 and 5 pesticides were found. In 18 bottles (21%), more than 6 up to 10 pesticides were present, and 5

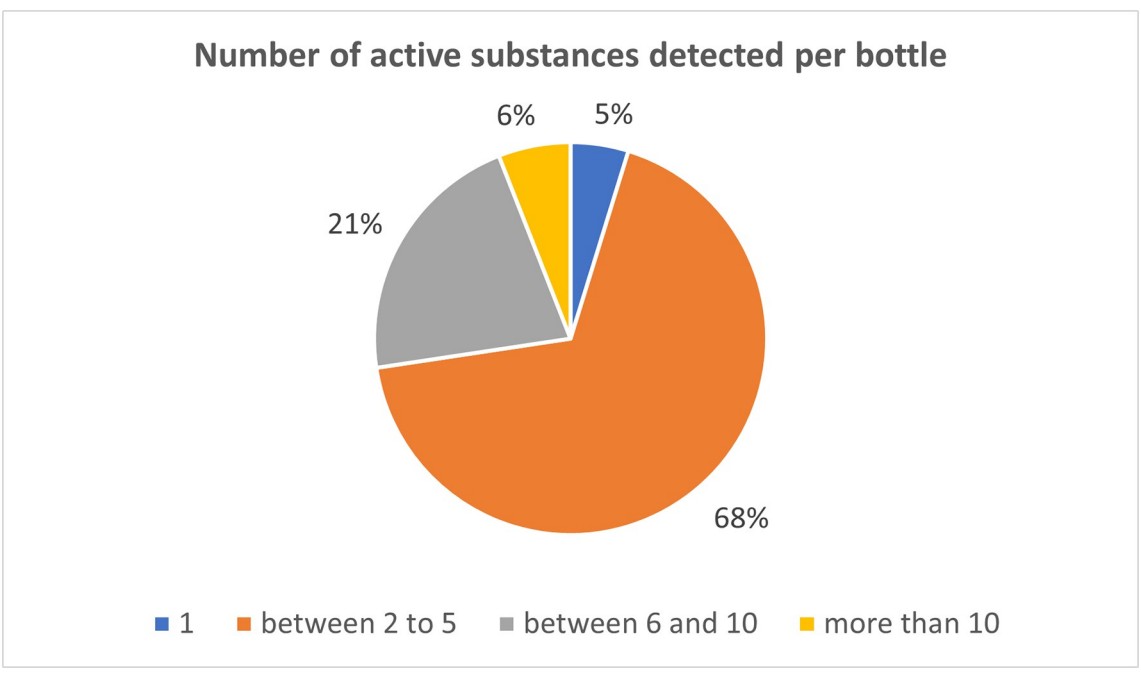

**Fig 5. The amount of pesticides detected in the analyzed wine, reported per wine bottle (N = 84).**

bottles (6%) even had more than 10 different pesticides detected. Notably, six bottles contained traces of 10 or more substances. It is worth mentioning that these six bottles with traces of more than 10 pesticides were all from the years 1995, 1996, 1997, 1998, and 2000. Within the scope of our study, these wines are considered relatively young. In this study our analytical method was not able to include and quantify obsolete and banned pesticides such as DDT and metabolites, aldrin and other POPs. Hence residues of these in the (old) wine may have been missed.

Fig 6 represents the relationship between age and the average number of pesticides detected using a point cloud. In this figure, the 2 wines from 1937 containing piperonylbutoxide are represented by two points that lie far outside the point cloud. As extreme values, they leverage the trend line and influence the result excessively. As can be observed from the graph there is a clear upward trend between the number of pesticides detected per bottle and the age of the wine. The younger the wine, the more traces of pesticides are found in the bottle.

Among the 21 pesticides detected, eight were found at concentrations exceeding 0.01 mg/L, specifically azoxystrobin, dimethomorph, metalaxyl, prochloraz, tebufenozide, pyrimethanil, carbaryl, and carbendazim. Carbendazim was the most frequently detected pesticide, present in eight wine bottles, while the remaining pesticides were identified in only one or two bottles. Three pesticides—tebufenozide, pyrimethanil, and carbaryl—were found at concentrations exceeding 0.05 mg/L (Table A6). Notably, carbaryl exhibited the highest concentration recorded in the entire study, observed in wine bottle F29. When compared to the official EU maximum residue levels (MRLs) for pesticides in wine, only carbaryl exceeded its permitted limit of 0.01 mg/L. The other detected pesticides remained below their respective regulatory thresholds, some of which are relatively high. For instance, the allowable limit for pyrimethanil is set at 5 mg/L (36). These findings indicate that while most pesticide residues in the tested wine samples were within acceptable limits, carbaryl represents an exception, exceeding EU regulatory standards.

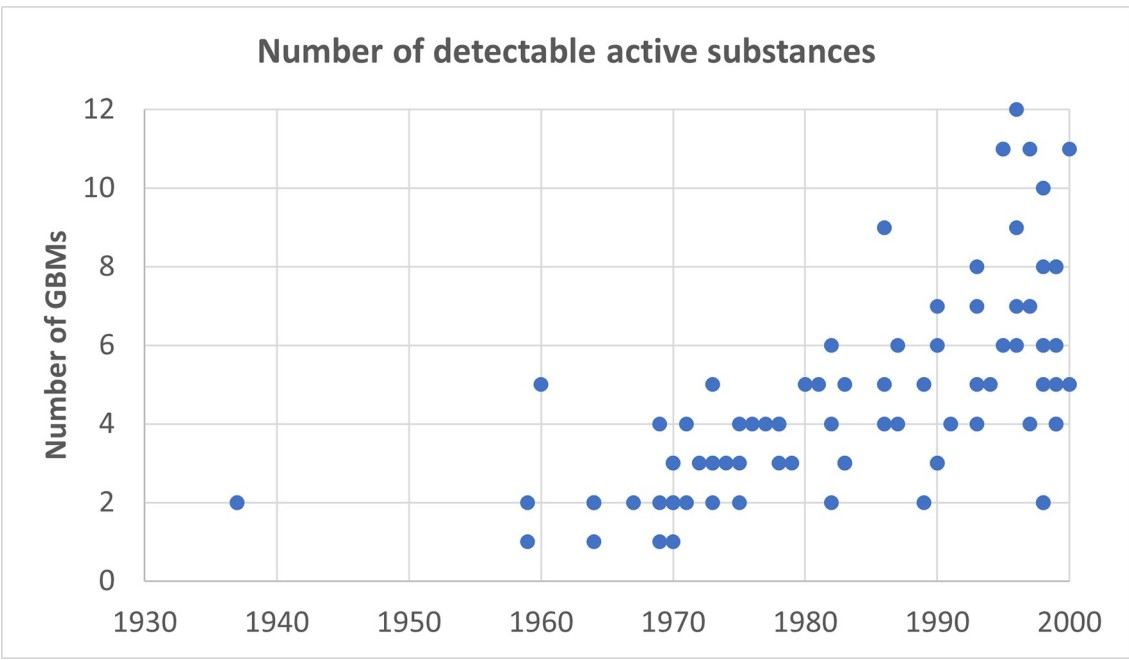

**Fig 6. The relationship between the production year of the analyzed wine and the number of detectable pesticides present in the wine.**

Several assumptions can be made regarding the apparent increase in the number of pesticides found in wine over the years. The most apparent explanation is that there was, in fact, less use of PPPs in the earlier years. However, there are other possibilities to consider. The analysis method did not include the old organochlorine pesticides that might have been missed. Also, the shift from broad-spectrum PPPs to selective ones since the 1990s may explain this outcome. It became evident that broad-spectrum PPPs also affected non-target organisms. Consequently, research began to focus on selective PPPs, aiming to preserve beneficial control agents. The consequence of this trend was that more PPPs were required to achieve the same level of crop protection. Similarly, a noticeable increase in the research field of pesticides has prompted their increased usage in food production.

Another explanation could be that older wines contain PPPs that require a different single-component detection technique, which might not have been detected using the extraction and detection methodology applied in this study. Older bottles may also be in a more advanced state of degradation, potentially influenced by lower alcohol content. It's important to note that there is a considerable time gap between the youngest and oldest wines studied. The varying half-life degradation times of different PPPs might have influenced the persistence of PPP residues in the bottles.

**3.1.3. PPPs detected in analyzed wines according to region.** S1 Table provides information about the grape varieties used in conjunction with specific wines. This data illustrates that certain grape varieties are frequently associated with particular wine regions. For example, Pinot Noir is consistently found in the Burgundy region, while Cabernet Franc, Merlot, and Cabernet Sauvignon are commonly linked to the Bordeaux region. However, it is evident that these grape varieties are not exclusively limited to one region, and there is overlap between them. For instance, the Cabernet Franc variety is also used in the Loire region. Fig 7 illustrates that the average number of pesticides per bottle of wine does not exhibit significant differences

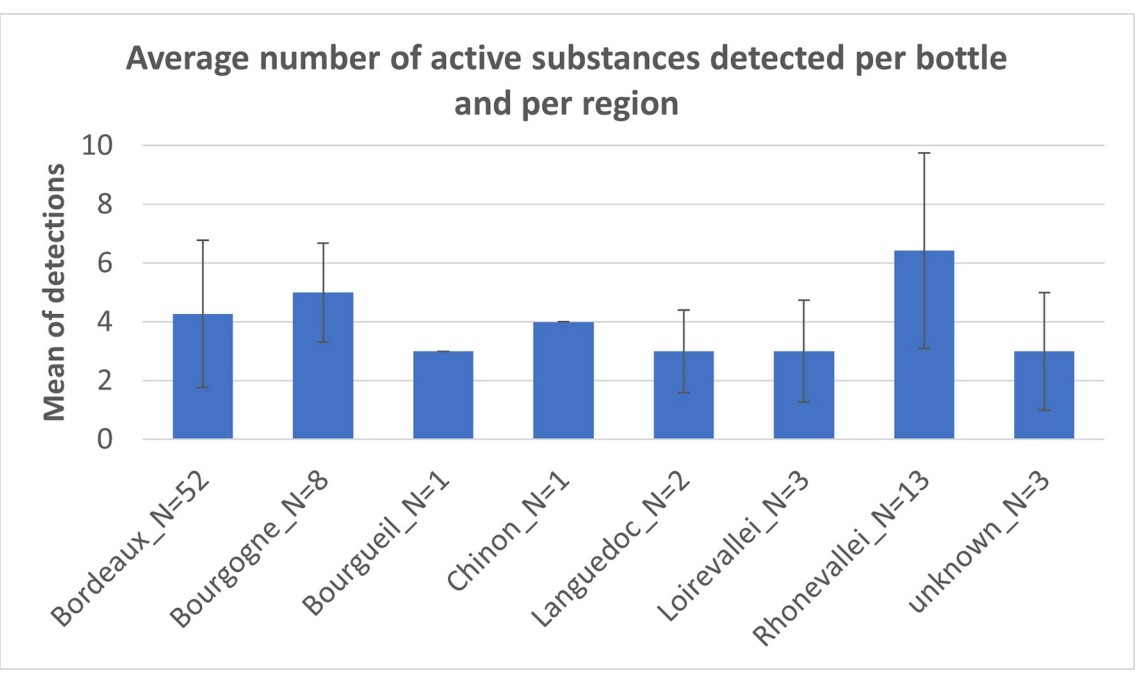

**Fig 7. The average number of pesticides detected per wine bottle, categorized by region.**

among various wine regions in France. Therefore, no further conclusions can be drawn based on region in relation to the use of PPPs.

**3.1.4. PPPs detected in analyzed wines according to authorization.** A noteworthy observation regarding the presence of PPPs in the wine, is the differentiation between pesticides that are still approved and those that are no longer approved by the European Commission (EC) as of 2022. Out of the 23 detected PPPs, 13 are no longer approved, while 7 agents remain approved for use in vineyards. Fig 8 highlights that there are still a considerable number of plant protection products in wines that have not received approval from the EC. On average, each bottle of wine contains traces of 2.9 non-approved agents compared to traces of 1.5 approved agents. These findings are intriguing, as they indicate a higher presence of unapproved pesticides in older wines. It is important, however, to note that these results are specific to the analyzed wines. Additionally, at the time of production, many pesticides were not restricted in the market. Moreover, the concentrations at which these substances are detected (>LOQ or >LOQ and >LOD) do not differ significantly between approved and unapproved agents.

## 3.2. PPPs detected in wine sediment

The results of the analyses of the wine sediment for each bottle can be found in S7 Table.

A total of 11 pesticides were detected in the wine sediment at concentrations exceeding 0.01 mg/L. These pesticides included cadusafos, carbaryl, carbendazim, diethofenocarb, difenoconazole, dimethomorph, metalaxyl, pyrimethanil, tebuconazole, azoxystrobin, and tebufenozide. Pesticides were detected more frequently in the sediment compared to the liquid phase of the wine. For instance, carbendazim was detected in the sediment of 40 wine bottles, while cadusafos and pyrimethanil were each identified in 9 bottles, and carbaryl in 6 bottles. Several pesticides were detected at concentrations exceeding 0.05 mg/L, which was more common in the sediment than in the liquid phase (Table A7). Of the 40 detections of carbendazim, 38 were

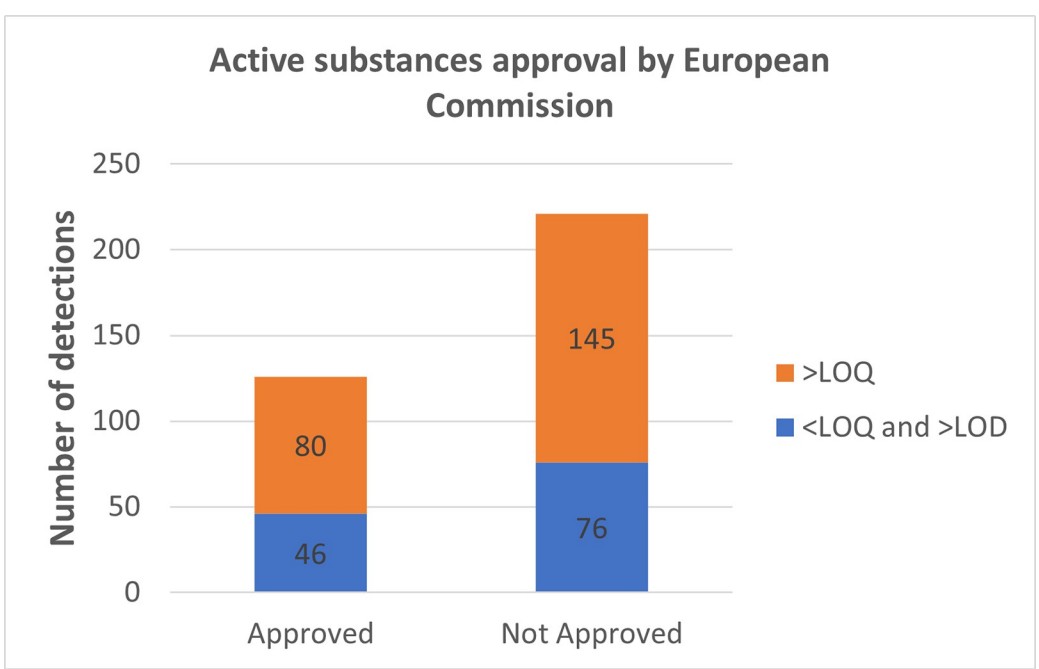

**Fig 8. The distribution of the average number of pesticides per bottle in the analyzed wines, categorized based on their approval status by the European Commission in 2022.**

above 0.05 mg/L, with the highest concentration, 9.37 mg/L, recorded in bottle F29. This value significantly exceeds the EU regulatory limit of 0.05 mg/L.

Additionally, other pesticides, such as cadusafos, carbaryl, carbendazim, and diethofenocarb, were detected at concentrations exceeding the EU maximum residue levels (MRLs). While a greater diversity of pesticides was found in the liquid phase overall, only carbaryl exceeded the allowable limits in that phase. Furthermore, the detection of pesticides in the liquid and sediment phases did not consistently occur in the same bottles. This highlights the significant accumulation of certain pesticides in the sediment, often at levels exceeding regulatory limits, emphasizing the importance of monitoring pesticide residues in both wine phases. A significant difference is also observed when comparing the number of active ingredients in the wine sediment per bottle to the wine phase. In the wine sediment, notably fewer pesticides per bottle were detected, with a maximum of only 5 pesticides per bottle (Fig 9), while in the wine, this number reached as high as 12. As indicated in S9 Table, in the wine sediment, an average of 0.8 pesticides were detectable below the LOQ, and an average of 1 pesticide could be quantified (>LOQ). Consequently, on average, there were 1.7 pesticides present per bottle in the sediment. Interestingly, in 15% of the sediment samples, no pesticides were detected, whereas in each wine sample, at least 1 pesticide was present. The lower pesticide levels in the wine sediments as compared to the wine liquid phase–even though many of the fungicides exhibit high sorption coefficients–may also be due to the ethanol content in the wine, as the ethanol increases the pesticides' solubility in the wine liquid. In contrast to the wine, there was no clear correlation between the year of the bottle and the number of active ingredients present per bottle in the wine sediment. The relationship between the year and the number of active ingredients per bottle in the wine sediment remained relatively consistent over the years (as shown in Fig 10).

Interestingly, only 12 pesticides are detected in the wine sediment phase, while 20 substances are detected in the wine liquid phase. Out of the 12 detected substances in the

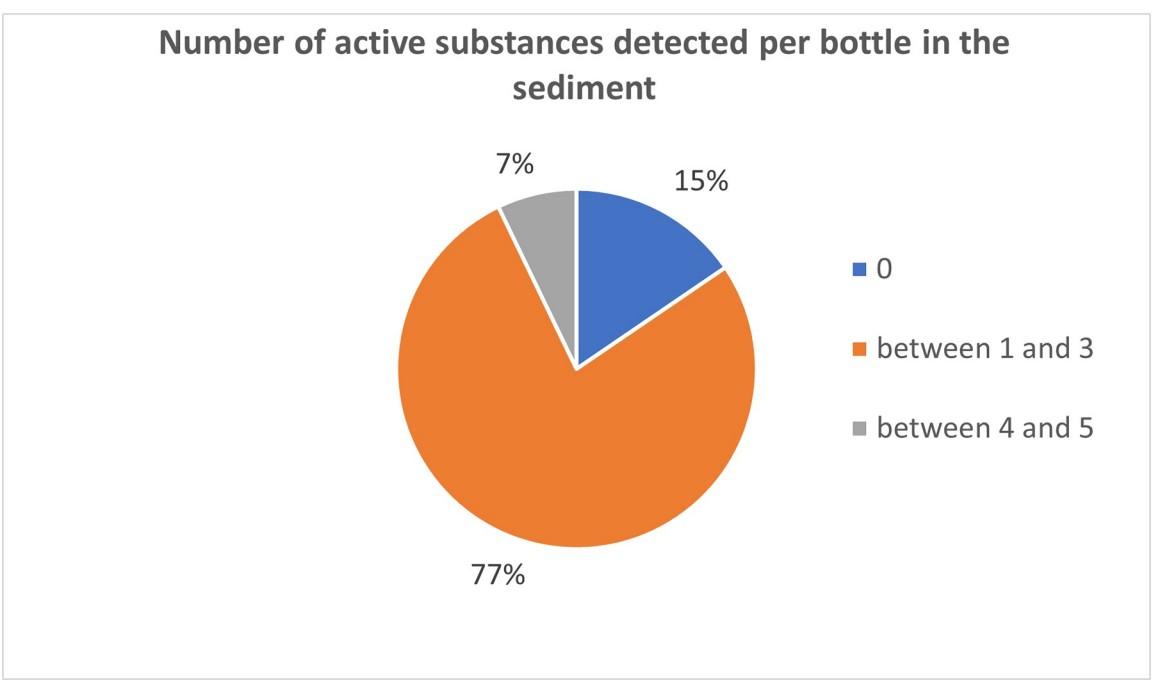

**Fig 9. The number of pesticides detected in the wine sediment per wine bottle, based on the analysis of 84 bottles (N = 84).**

sediment, 11 were found at concentrations above the LOQ. Atrazine appears to be the only substance that is exclusively present in the sediment and not in the wine itself. Notably, the two most frequently detected substances in both, wine and wine sediment, were carbendazim

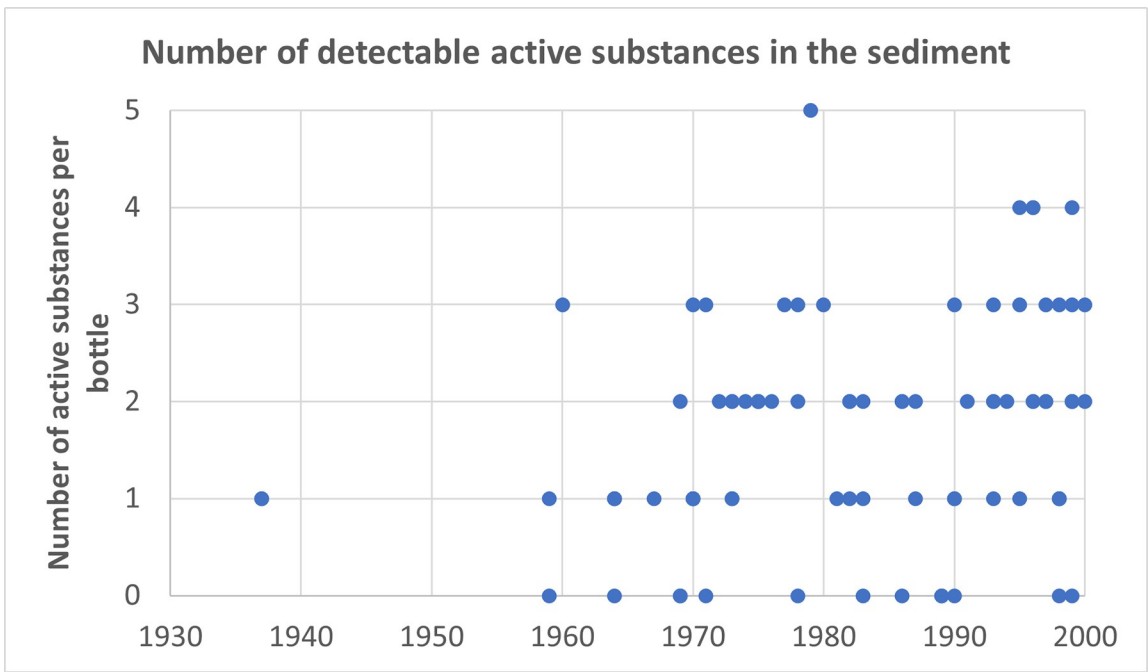

**Fig 10. The relationship between the production year of the wine and the number of detectable pesticides found in the wine sediment.**

and cadusafos. It's worth mentioning that cadusafos could be quantified nine times in the sediment, while it was not quantifiable above LOQ in the wine liquid. This suggests a preference for cadusafos to be present in the wine sediment rather than the wine itself.

When comparing the average quantifiable concentrations between the wine sediment and wine, it was observed that concentrations of the PPPs in the sediment were generally much higher. This is because non-polar pesticides exhibit hydrophobic characteristics and therefore have a lower affinity for the liquid.

## 3.3. Copper in old wines

Copper has historically been, and in some cases continues to be, employed in vineyards to combat fungal diseases. Excessive copper intake can lead to potential health problems. The World Health Organization (WHO), in conjunction with the European Union (EU), has established the maximum permissible standard for copper in drinking water at 2 mg/L [30]. Maximum residue limit (MRL) value for wine is 1 mg of Cu per L (EEC regulation No. 606/2009, Annex 1 A). S10 Table presents the results of copper analysis for the same samples from the 84 wines. It is worth noting that the copper standard was not exceeded in all samples, with the exception of three. The highest recorded value, twice above the norm, was 2.080 mg/L for sample 81. Two other samples contained just over of the allowed amount, measuring 1.308 mg/L and 1.154 mg/L for samples 83 and 84, respectively.

## 3.4. Food safety

S1 Table summarizes the maximum residue values found in the analyzed wine bottles for 20 pesticides, with no traces of cadusafos detected. Exposure was calculated for both female and male consumers, based on their frequency of wine consumption. This approach provides a high-risk estimate for the presence of active ingredients in analyzed wines.

From these findings, it can be concluded that the Acceptable Daily Intake (ADI) was not exceeded for all pesticides except carbaryl. A noteworthy result was observed in sample F29, a 1969 Bordeaux known as Château Bries-Caillou (refer to S1 Table). Carbaryl concentration exhibited an exceptionally high residue of 2.95 mg per 1L of wine (see S11 Table). The exposure calculation for female drinkers in Belgium resulted in values of 0.013 and 0.029 mg per kg of body weight per day for average and frequent consumption, respectively. These values significantly surpass the ADI for carbaryl, which is 0.0075 mg/(kgBWd). Overall, the low ADI value indicates the presence of more toxic substances, suggesting that a pesticide was present in wine samples at levels exceeding safe toxicity thresholds by 77% and 285%. A similar calculation for male average and frequent drinkers yielded values of 0.011 and 0.024 mg/(kgBWd), respectively, also exceeding the ADI for carbaryl by 49% and 225%. S11 Table also compares female and male exposure to the Acute Reference Dose (ARfD), revealing that carbaryl toxicity was significantly exceeded for both genders, regardless of their wine consumption frequency.

The most stringent ADI (0.0004 mg/(kgBWd)) is established for cadusafos. Residues of cadusafos were found in 34 wine samples, but none exceeded the LOQ, making quantification impossible. However, it can be deduced that a concentration of 0.1825 mg/L of wine would be necessary to achieve an exposure of 0.0004 mg/(kgBWd). This concentration of 0.0004 mg is still well above the LOQ of 7.51E-5 mg/L of wine. As no cadusafos residue exceeded the LOQ, no safety concerns were raised.

Most pesticides are organic molecules. Pesticide degradation adheres to a first-order kinetic model, characterized by an exponential decrease in concentration over time. This model suggests that the more the pesticide levels diminish, the longer it takes to further degrade. Pesticide degradation may also be bi-phasic, e.g. fast degradation of dissolved pesticide with a slow

degradation of sorbed pesticide, It also depends on the system/matrix. A notable finding from our study is that, thanks to the high sensitivity of our analytical techniques, these pesticides can be detected even after a minimum of 20 years of storage. Moreover, the presence of alcohol plays an important role in inhibiting microbial activity, which would otherwise accelerate degradation. Consequently, pesticides dissolved in the alcohol phase exhibit greater stability and persistence compared to those present in water. This results in pesticides being more effectively preserved in alcoholic beverages than in other types of food and drink.

This risk assessment does not take into account the cumulative exposure to pesticides. The risks associated with pesticide residues in food are typically evaluated on a substance-by-substance basis in wine. However, many pesticides exhibit similar effects, and their combined impact on human health may be greater than when considered individually. Additionally, consumers do not limit their intake to wine alone; residues of the same pesticides may also be present in other food items consumed throughout the day.

In Belgium, the average wine consumption stands at 300g per day, while frequent wine consumers consume 653g per day [24]. Consequently, the average Belgian ingests approximately 29.6g of ethanol daily, while frequent consumers ingest more than double this amount, equating to 64.41g of ethanol per day. Considering the grams per Body Weight (BW) statistics, Belgians consume 10g of wine per unit of body weight daily, translating to 1g of ethanol per kg of body weight ingested. It is well-established that chronic ethanol intake can lead to various diseases that subsequently impact lifespan. The most common consequence of excessive ethanol consumption, known as alcohol intoxication, is liver problems, alongside other metabolic issues [31]. Thus, when consuming wine, the primary health risks are more likely to arise from ethanol presence rather than pesticide contamination.

The generally accepted 'safe' copper concentration in wine is less than 0.3–0.5 μg ml−1 [32]. While many studies indicate that moderate wine consumption does not pose significant health risks related to copper [33,34], but all these findings are based on average wine exposure. Increased wine consumption, particularly among wine connoisseurs or collectors of old wines, may present toxicological risks due to limited public awareness about potentially banned ingredients. Prolonged consumption of copper in wine has been associated with various health issues, including respiratory diseases, acute gastritis, and decreased antioxidant capacity [35]. Therefore, it is essential to expand consumer knowledge not only about pesticide residues but also about copper content in wine.

Maximum Residue Limit (MRL) legislation varies across regions, with distinct MRL values established for different pesticides. Importantly, these limits are typically set for grapes rather than the final wine product. Pesticides not approved for use in grape production have default MRL values set at 0.01 mg/kg or 0.05 mg/kg. To determine MRLs for wine, a processing factor is applied, which adjusts for changes in pesticide residue levels during the winemaking process. This factor accounts for how residues may increase or decrease through stages like washing, crushing, fermentation, and filtration, thereby ensuring that the final wine product meets safety standards. We refer for details on the MRLs of wine grapes to the EC official website [36].

Considering the quantity and variety of pesticides detected in these older wines, products having residues above the default values 0.01 or 0.05 mg/kg are not allowed anymore for use on grapes, as those fail to meet current European regulatory standards. These bottles should normally be taken out of the market. This raises a compelling question: why do consumers perceive and approach their exposure to pesticides differently when consuming aged wines compared to other agricultural commodities? In the latter case, products are frequently verified and removed from the market based on changes in legislation regarding pesticide content following novel findings on pesticides when consumed or present in the environment.

Consequently, aged wines might be better classified as collector's items due to their pesticide content rather than as everyday beverages. This intriguing contrast in consumer behavior and perception prompts further examination. It initiates a discussion about the factors influencing consumer attitudes and choices concerning pesticide exposure in various contexts. It also underscores the need for a deeper understanding of these dynamics.

## 4 Conclusion

The study's findings shed light on the persistent presence of unauthorized substances, as well as copper residues, in aged wines. The analysis method here did not include the old organo-chlorine pesticides that might have been missed. Despite not focusing on cumulative pesticide exposure, it uncovered trace amounts of 21 different pesticides derived from Plant Protection Products (PPPs) [7]. These substances, though found in small concentrations, underscore a broader issue of contamination in vintage wines. While most of the residue levels identified in the samples were deemed non-threatening to human health, one sample in wine revealed an alarmingly high concentration of carbaryl, a pesticide that exceeded toxic consumption thresholds. Five samples for wine sediment had also levels of pesticides exceeding allowed norms. Although, when enjoying wine, it can be assumed, for the most part, that the wine is not shaken, and only the liquid part is consumed, leaving the wine sediment in the glass or bottle.

The presence of such contaminants highlights the need for more rigorous pesticide testing across a wide array of products, including those that, like vintage wines, are often overlooked due to their status as collectibles or luxury items. Currently, stringent food safety regulations are in place for modern products, but aged wines and other historical consumables often slip through regulatory cracks. This study underscores the necessity of extending contemporary testing protocols to these older items, ensuring that even vintage wines, which may have been produced under different regulatory frameworks, are subject to the same safety scrutiny.

A key point of this research is that it does not seek to address whether the pesticides found in the analyzed wines were legal at the time they were produced. As regulations evolve, substances that were once widely used in agriculture may now be banned, leading to potential risks from historical contamination. Consumers may not consider this possibility of contamination, assuming these wines are completely safe to drink. Luckily most of the investigated wines are still acceptable for the market. However, this research suggests that they might be held to the same standards as other food and drink products currently on the market. By bringing attention to this overlooked issue of pesticides authorised previously in time, the study encourages further discussion on how to handle the sale and consumption of such products, especially in light of modern food safety regulations that no longer permit these substances.

## Supporting information

**S1 Table. A list of tested French wines, including sample codes, growing regions, and years of production.**
(DOCX)

**S2 Table. Percentage of recovery for active ingredients for wine extraction.**
(DOCX)

**S3 Table. Percentage recovery of active ingredients for wine sediment extraction.**
(DOCX)

**S4 Table. A summary of different pesticides detected in wine, with concentrations exceeding the LOQ.** Column 2 -the percentage of samples with concentrations higher than LOQ.

Column 3—the average concentration of each pesticide across all samples with levels above the LOQ excluding the zero samples. Column 4 –ADI values. Column 5—the release year for each pesticide, respectively.
(DOCX)

**S5 Table. Pesticides detected in wine (1st column).** Column 2—the type of PPP. Column 3-the main target organism for each pesticide. Column 4 -the mode of action for each PPP (systemic and contact). Column 5—the MRL for grapes as per the latest European Commission regulations. Column 6—indicates whether the pesticide is authorized for use in the European Union as of 2022, according to the European Commission laws [37].
(DOCX)

**S6 Table. The residues (mg/L) of pesticides detected liquid phase in wine samples (in green values above 0.01mg/L, in yellow values above 0,05mg/L).**
(DOCX)

**S7 Table. The residues (mg/L) of pesticides detected in the wine sediment of wine samples (in green values above 0.01mg/L, in yellow values above 0,05mg/L).**
(DOCX)

**S8 Table. Amount of pesticides found per bottle of wine in the liquid phase (N = 84).**
(DOCX)

**S9 Table. The amount of pesticides found per bottle of wine in the wine sediment.**
(DOCX)

**S10 Table. The copper concentrations detected, per wine sample (mg/L).**
(DOCX)

**S11 Table. The maximum residue values found in one analyzed bottle compared to ADI values and ARfD values for each pesticide (average weight female: 66.7kg. average weight male: 99kg).**
(DOCX)

**S12 Table. List of pesticides used and validated in the analytical method, and monitored across all wine samples.**
(DOCX)

## Author Contributions

**Conceptualization:** Pieter Spanoghe.

**Formal analysis:** Hannah Vanderstappen, Lotte Neckebroeck, Dries Verhaegen, Pauline van den Hove.

**Investigation:** Jasmine De Rop, Lilian Goeteyn, Hannah Vanderstappen, Lotte Neckebroeck, Dries Verhaegen, Pauline van den Hove, Joachim Neri.

**Methodology:** Pieter Spanoghe.

**Supervision:** Pieter Spanoghe, Erik Meers.

**Writing – original draft:** Hannah Vanderstappen, Lotte Neckebroeck, Dries Verhaegen, Pauline van den Hove.

**Writing – review & editing:** Pieter Spanoghe, Agnieszka Deja-Muylle, Erik Meers.

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
