## [Decision Letter · Decision Letter 0]

8 Aug 2024

PONE-D-24-17146The temporal variation in pesticide concentrations within matured French wines.PLOS ONE

Dear Dr. Spanoghe,

Thank you for submitting your manuscript to PLOS ONE. After careful consideration, we feel that it has merit but does not fully meet PLOS ONE’s publication criteria as it currently stands. Therefore, we invite you to submit a revised version of the manuscript that addresses the points raised during the review process.

We look forward to receiving your revised manuscript.

Kind regards,

Totan Adak

Academic Editor

PLOS ONE

Journal Requirements:

4. Please remove your figures from within your manuscript file, leaving only the individual TIFF/EPS image files, uploaded separately. These will be automatically included in the reviewers’ PDF.

Additional Editor Comments:

I am extremely sorry for the delay. I was expecting few more reviews. I have gone through the comments of the reviewer. I found the suggestions will improve the manuscript. Please incorporate all the suggestions in your revised manuscript.

In addition, I have few more comments which should be addressed in your revised manuscript:

1. Method validation is a issue in this manuscript. Please follow SANTE guidelines. Otherwise, the manuscript will not be considered.

2. Choice of pesticides need to be elaborated. Why have you restricted to few compounds?

3. Please check the food safety calculation as suggested by the reviewer.

Reviewers' comments:

Reviewer's Responses to Questions

**Comments to the Author**

1. Is the manuscript technically sound, and do the data support the conclusions?

Reviewer #1: Partly

2. Has the statistical analysis been performed appropriately and rigorously? 

Reviewer #1: Yes

3. Have the authors made all data underlying the findings in their manuscript fully available?

Reviewer #1: Yes

4. Is the manuscript presented in an intelligible fashion and written in standard English?

Reviewer #1: Yes

5. Review Comments to the Author

Reviewer #1: Manuscript Number: PONE-D-24-17146

Article Type: Research Article

Full Title: The temporal variation in pesticide concentrations within matured French wines.

The manuscript entitled “The temporal variation in pesticide concentrations within matured French wines” addresses an important food safety issue associated with pesticide residues in wine. The authors presented the data very well, however some important key issues need to be addressed before accepting the manuscript for publication. The details are as given below;

Introduction:

Some of the key points may be discussed in introduction;

- Status of MRLs of pesticide residues in wine and wine grapes. The importance of processing factor/transfer rate and their relation to wine MRLs

- A brief review on analytical methods followed for pesticide residue analysis in wine may be discussed

- The authors made a detailed risk assessment of positive detection, but introduction does not see any such insight.

- The number of pesticides included in the study may be provided in the objective.

2 Material and methods:

The materials and methods should address the following;

- The year of sampling should be provided

- The chemicals and reagents used in the study along with their source; specifically, the pesticide standards (CRMs) used in the study should be provided.

- Did the authors analyzed red wine and white wine separately, are they used same procedure for both wine, is there any difference with respect to LOD and LOQ?

- The recovery specified for many pesticides studied are less than 50%., Did the authors applied recovery correction factor in calculating the residues? Why this much low recovery was observed? Did the authors calculate matrix effect? Is this low recovery being due to matrix suppression?

- What is the spiking concentrations where the authors did the recovery study? Which are the concentrations spiked for recovery study? One recovery data should necessarily be at LOQ level.

- How the authors established the LOD and LOQ for each analytes? Why the matrix effect did not calculate? How the quantification was performed in LC-MS, Matrix match calibration, or matrix based calibration or solvent calibration? These points should be included.

- Approximately, what is weight of sediments observed in the 50 mL wine? How much dilution factor applied for extraction in 15 mL CAN. Should be provided.

- Table A.2 and Table A.3 may be provided with retention time and mass transitions (Quantifier and Qualifier). All these pesticides are only tested or detected? If detected, should give detailed data about the list of pesticides tested.

(Overall, there is a lack of required information on analytical method followed, need further detailed information on analytical protocols as detailed above).

3.Results and discussion:

3.1.1. PPPs detected in old wines:

3.1.2. Number of PPPs detected in analyzed wine samples:

- How the classification of sample has been performed to old wines? It’s confusing what is old wines, analyzed wine, Are these data overlapping??

- The authors are made a detailed classification of the results based on plant protection products. However, some of the meaningful insight is missing such as; are these pesticides detected where officially allowed to use at the time of wine preparation?

- If any residue presents in old wine, what is the persistence of that pesticides in wine? Can we correlate that with persistence of pesticides?

- Authors may have a detailed discussion on MRLs of pesticide in wine and its compliance in tested wine.

- As per the results, less number of pesticides are detected in sediments which is ideally not expected since, the residues of pesticides will be more adsorbed in solid particles especially to the leese. Hence, this result may please be explained based on solubility of the pesticides in water/wine and octanol/water coefficient (log Kow values).

- The study is only focused on Belgian population and pesticides registered in Belgium is only considered. Is this is aligned with EU regulation? As per mu understanding, the list of pesticides covered under this study does not cover most of the pesticides used in EU countries. Is this work has only local importance?

-3.3. Copper in old wines: I the limit regulated in water is applicable to wine?! Discussion may be aligned with wine only. If water standard applicable to wine, the other pesticides limit in water also applicable to wine. Please consider this

3.4. Food safety:

- Only comparing the exposure data with ADI the food safety of the wine cannot be compared. A detailed dietary risk assessment of Belgian population based on dietary intake of other food commodities and their MRL status with respect to detected pesticide only will give a risk assessment since the residue can enter in to body through other food also. This should be considered while concluding the risk assessment. The limitation should be clearly specified.

4 Conclusion: Considering the comment in 3.4; the conclusion may be modified with the limitation of the study.

6. PLOS authors have the option to publish the peer review history of their article (what does this mean?). If published, this will include your full peer review and any attached files.

Reviewer #1: **Yes: **Ahammed Shabeer TP

---

## [Author Response · Author response to Decision Letter 0]

23 Sep 2024

Introduction: 

Some of the key points may be discussed in introduction;

1. Status of MRLs of pesticide residues in wine and wine grapes. The importance of processing factor/transfer rate and their relation to wine MRLs

Following text added in the manuscript in line 72-84

2. A brief review on analytical methods followed for pesticide residue analysis in wine may be discussed 

Following text added in the manuscript in line 57-72

3. The authors made a detailed risk assessment of positive detection, but introduction does not see any such insight. Text considering this has been added to the manuscript line 95-97.

4. The number of pesticides included in the study may be provided in the objective. . Text considering this has been added to the manuscript. Line 92.

2 Material and methods:

The materials and methods should address the following; 

5. The chemicals and reagents used in the study along with their source; specifically, the pesticide standards (CRMs) used in the study should be provided. 

See line 156-158 text for standards have been added. Producers and sources of chemicals used in materials and methods are also added in the text. 

6. Did the authors analyzed red wine and white wine separately, are they used same procedure for both wine, is there any difference with respect to LOD and LOQ? 

Only red wine was analyzed and thus only one procedure was used. As a result we obtained one LOD and LOQ dataset.

7. The recovery specified for many pesticides studied are less than 50%., Did the authors applied recovery correction factor in calculating the residues? Why this much low recovery was observed? Did the authors calculate matrix effect? Is this low recovery being due to matrix suppression? 

 Yes, a recovery correction factor was applied to the results to account for the matrix effects, which were significant in this case. Despite these effects, the method demonstrated adequate reproducibility. To evaluate the matrix, organic wine was used, providing a representative sample for the analytical performance. A multi-residue method was employed to address the challenge that one method cannot extract all pesticides, with different polarity/solubility, in a single solvent, at 100% efficiency. This approach ensures comprehensive detection and quantification of various pesticides, compensating for the limitations inherent in solvent-based extraction methods.

8. What is the spiking concentrations where the authors did the recovery study? Which are the concentrations spiked for recovery study? One recovery data should necessarily be at LOQ level. PSA not explained?

10 µL of the pesticide was spiked in the concentration of 1 mg/L, which is a close to the level of quantification. PSA has been explained in the line 139.

9. How the authors established the LOD and LOQ for each analytes? Why the matrix effect did not calculate? How the quantification was performed in LC-MS, Matrix match calibration, or matrix based calibration or solvent calibration? These points should be included. 

The Limits of Detection (LOD) and Limits of Quantification (LOQ) were determined by injecting the lowest concentration eight times and by analyzing the signal-to-noise (S/N) ratio. The variability was assessed by repeating the measurements three times for the LOD and ten times for the LOQ. The quantification has been according to the standard curve. Quantification was performed based on pre and post spiking of the wine (pre-spiking of the wine, post – spiking of the wine extract). The fixed value used is based on the lowest standard concentration injected.

Line added in 179-184.

10. Approximately, what is weight of sediments observed in the 50 mL wine? How much dilution factor applied for extraction in 15 mL CAN. Should be provided. 

Average weight of the sediment is 2g. There was no dilution applied but up concentration. From 15ml mix 5ml was taken and finally 2 ml was analyzed (see method chapter). Every step of analysis has taken it into account further. 

11. Table A.2 and Table A.3 may be provided with retention time and mass transitions (Quantifier and Qualifier). All these pesticides are only tested or detected? If detected, should give detailed data about the list of pesticides tested. 

The pesticides were not detected but still they have been validated. That means that the analytical protocol was applied and checked on the pesticides added to the organic wine.

3.Results and discussion: 

3.1.1. PPPs detected in old wines:

3.1.2. Number of PPPs detected in analyzed wine samples:

12. How the classification of sample has been performed to old wines? It’s confusing what is old wines, analyzed wine, Are these data overlapping??

 The wine bottles were classified as "old" based on their production dates. However, there was some confusion between the terms "old" and "analyzed wine" in the naming conventions. These terms were previously used interchangeably, but this issue has now been addressed and corrected in the text.

13. The authors are made a detailed classification of the results based on plant protection products. However, some of the meaningful insight is missing such as; are these pesticides detected where officially allowed to use at the time of wine preparation? 

This research was not aimed to address whether the pesticides in question were permitted at the time the analyzed wines were produced. Instead, the primary objective was to highlight the need for re-evaluating collector's food products, such as vintage wines, against current, stringent food safety standards. Additionally, the aim was to increase public awareness about the potential pesticide residues in these older products. Contemporary food regulations do not permit residues of substances that are banned today, even if those substances were previously allowed. This study seeks to illuminate the issue of historical contamination and offers a new perspective on assessing wines compared to other food items. This statements have been incorporated in the conclusions.

14. If any residue presents in old wine, what is the persistence of that pesticides in wine? Can we correlate that with persistence of pesticides?

Pesticide degradation adheres to a first-order kinetic model, characterized by an exponential decrease in concentration over time. This model implies that as pesticide levels diminish, the time required for further degradation extends. A notable finding from our study is that, thanks to the high sensitivity of our analytical techniques, these pesticides can be detected even after a minimum of 20 years of storage. Moreover, the presence of alcohol plays an important role in inhibiting microbial activity, which would otherwise accelerate degradation. Consequently, pesticides dissolved in the alcohol phase exhibit greater stability and persistence compared to those present in water. This results in pesticides being more effectively preserved in alcoholic beverages than in other types of food and drink. This statements have been incorporated in the conclusions.

15. Authors may have a detailed discussion on MRLs of pesticide in wine and its compliance in tested wine. 

Indeed, Maximum Residue Limit (MRL) legislation varies by region, with distinct MRL values established for different pesticides. Importantly, these limits are typically set for grapes rather than the final wine product. Most pesticides are banned in grape production, and default MRL values are commonly set at 0.01 mg/kg or 0.05 mg/kg. During wine production, a processing factor is applied to account for pesticide residues in the final product. This issue has been discussed in detail previously; for more comprehensive information, please refer to the relevant paper. We refer for details on the MRLs of wine grapes to the EC official website. This website is regularly updated. https://ec.europa.eu/food/plant/pesticides/eu-pesticides-database/start/screen/products/details/38

This part was added to the manuscript in line 482-487.

16. As per the results, less number of pesticides are detected in sediments which is ideally not expected since, the residues of pesticides will be more adsorbed in solid particles especially to the leese. Hence, this result may please be explained based on solubility of the pesticides in water/wine and octanol/water coefficient (log Kow values). 

This comment is pointing to the an interesting point and we have tried to answer it as best as possible. We post a table summarising Kow values at the end of this comment. However, we must admit that there was no obvious relations ship between water solubility and the water/wine Kow. This can be explained, however, by the fact that we analyse 3 matrixes, namely: residue in water, residue in the alcohol and finally in the sediment. The final outcome is a distribution of the three. Hence, the table with no clear conclusions have not been added to the paper manuscript. 

 17. The study is only focused on Belgian population and pesticides registered in Belgium is only considered. Is this is aligned with EU regulation? As per my understanding, the list of pesticides covered under this study does not cover most of the pesticides used in EU countries. Is this work has only local importance? 

This study is locally significant as it exclusively examines wine bottle samples from Belgian collections. However, the issue of old collector wines containing substances that no longer comply with current health standards is likely to be widespread, particularly within the European Union (EU), where food regulations are harmonized across member states. While the consumption data pertains specifically to Belgian consumers, the residue legislation is regulated at the European level, as outlined in the MRL chapter. Thus, this study is well-aligned with EU regulations.

Moreover, all analyzed wines originated from France, and our research focuses on wine production and pesticide use in that country. Given that France has historically exported wine not only to Belgium but also internationally, it is plausible that older wines stored in other countries may also contain pesticides no longer permitted for wine grape production in Europe. Consequently, such wines are likely to be non-compliant with current EU food laws. This part was also incorporated in the summary.

3.3. Copper in old wines: 

18. I the limit regulated in water is applicable to wine?! Discussion may be aligned with wine only. If water standard applicable to wine, the other pesticides limit in water also applicable to wine. Please consider this. 

Discussion based on literature study has been added in lines 473-480. Wine standard for copper has been added. 

3.4. Food safety: 

19. Only comparing the exposure data with ADI the food safety of the wine cannot be compared. A detailed dietary risk assessment of Belgian population based on dietary intake of other food commodities and their MRL status with respect to detected pesticide only will give a risk assessment since the residue can enter in to body through other food also. This should be considered while concluding the risk assessment. The limitation should be clearly specified. 

We added this point inside the text in line 455-461

4 Conclusion: Considering the comment in 3.4; the conclusion may be modified with the limitation of the study. 

This risk assessment does not consider cumulative exposure of pesticides.

Table 1: Solubility for all pesticides detected in wine and all pesticides in sediment (source: https://sitem.herts.ac.uk/aeru/ppdb/)

Table 2: List of pesticides that were used and validated in the analytical method and monitored in all wine bottles

Acephate Atrazine fenpyroximat 

Acetamiprid Azoxystrobine carbetamide 

Amethryn Benalaxyl carbosulfan 

Boscalid Bentazon carfentrazon-ethyl

Butachlor Bitertanol chlorprofam 

Carbaryl Cadusafos clopyralid 

Carbofuran Carbendazim cyprodinil 

Difenconazole Chlorotoluron diethofencarb 

Dimethoate Chlorpyrifos difenacoum 

Dimethomorph Cyanizine diflubenzuron 

Diuron Cyflufenamid flazasulfuron 

Ethoprophos Cymoxanil fluazufop-p-butyl

Fenamiphos Diazinon flufenacet 

Fenbuconazole Epoxiconazole indoxacarb 

Fenpropimorf Fenoxycarb metamitron 

Fludioxonil Hexythiazox metazochlor 

Hexaconazole Imazalil methabenzthiazuron

Malathion Imidacloprid methoxifenozide

Metalaxyl Iprodione piperonylbutoxide

Methiocarb Isoproturon prosulfocarb 

Methomyl Kresoxim-methyl pyridaben 

Methribuzin Linuron sethoxydim 

Monocrotophos Metsulfuron-methyl tau fluvalinate 

Pendimehtanil Nicosulfuron triadimefon 

Pirimicarb Oxamyl triticonazole 

Prochloraz Parathion zoxamide 

Propoxur Penconazole Terufos 

Prosulfocarb Pirimiphos-methyl Thiacloprid 

Pyrazosulfuron-ethyl Profenofos Triademinol 

Pyrimethanil Propanil Thiofanate-methyl 

Spinosad a Propazine 

Spinosad d Propiconazole 

Spiroxamine Pyrachlostrobine 

Tebuthiuron Simazine 

Thiabendazole Spirodiclofen 

Thiametoxam Tebuconazole 

Thifensulfuron-methyl Tebufenozide 

Thiodicarb Temephos 

Tirazophos Terbutryn 

Trifloxystrobine Terbutylazine

---

## [Decision Letter · Decision Letter 1]

14 Oct 2024

PONE-D-24-17146R1The temporal variation in pesticide concentrations within matured French wines.PLOS ONE

Dear Dr. Spanoghe,

Thank you for submitting your manuscript to PLOS ONE. After careful consideration, we feel that it has merit but does not fully meet PLOS ONE’s publication criteria as it currently stands. Therefore, we invite you to submit a revised version of the manuscript that addresses the points raised during the review process.

We look forward to receiving your revised manuscript.

Kind regards,

Totan Adak

Academic Editor

PLOS ONE

Journal Requirements:

Reviewers' comments:

Reviewer's Responses to Questions

**Comments to the Author**

1. If the authors have adequately addressed your comments raised in a previous round of review and you feel that this manuscript is now acceptable for publication, you may indicate that here to bypass the “Comments to the Author” section, enter your conflict of interest statement in the “Confidential to Editor” section, and submit your "Accept" recommendation.

Reviewer #1: (No Response)

Reviewer #2: (No Response)

2. Is the manuscript technically sound, and do the data support the conclusions?

Reviewer #1: Yes

Reviewer #2: Partly

3. Has the statistical analysis been performed appropriately and rigorously? 

Reviewer #1: Yes

Reviewer #2: Yes

4. Have the authors made all data underlying the findings in their manuscript fully available?

Reviewer #1: Yes

Reviewer #2: Yes

5. Is the manuscript presented in an intelligible fashion and written in standard English?

Reviewer #1: Yes

Reviewer #2: No

6. Review Comments to the Author

Reviewer #1: The authors has addressed most of the comments raised in the first revision. However, few critical issues need to be incorporated as detailed below;

-The authors not specified the list of pesticides monitored under this study.

-The recovery data provided for wine and sediments, the list of pesticides are different.

-The authors should include a separate paragraph on "Analytical method performance and quality control" under "Results and discussion section. This should include the LOD, LOQ and recovery of all the pesticides considered in monitoring.

Reviewer #2: It is a g good idea to screen bottled aged wine for pesticide residues. The range of pesticides analyzed is fair, which adds to the conclusions that can be drawn. Results on the pesticide levels found could be discussed more. The paper focuses more on the pesticides detected than their actual levels, and more focus could be put on the levels detected. A clear language is lacking here and there. Material & Method section needs more work. See all my comments in the separate Word file.

7. PLOS authors have the option to publish the peer review history of their article (what does this mean?). If published, this will include your full peer review and any attached files.

Reviewer #1: No

Reviewer #2: No

---

## [Author Response · Author response to Decision Letter 1]

2 Dec 2024

Dear reviewer, we thank you for reading our manuscript carefully and for the comments that shed a new light on some of the inconsitencies of the text. Please find our answers below and a manucript version with track changes.

Manuscript Number PONE-D-24-17146R1

The temporal variation in pesticide concentrations within matured French wines.

General comments from reviewer:

-It is a g good idea to screen bottled wine for pesticide residues. The range of pesticides analyzed is fair, which adds to the conclusions that can be drawn (although I don’t have a very good overview of pesticides approved in grapes), but the pesticide list need to be clarified better (e.g. in Appendix). Has been added to the appendix

-It can be confusing when you refer to «active substance» in figures, but use «PPP» in the text. The «PPP» should in most cases be understood as «active substance in PPP». A PPP is the formulated product, that contains one or more active substances and co-formulants. You refer to [12] Sykalia et al (2024), and they use the term «Pesticide» for all the active substances in their paper – as is also commonly used in other papers. Although «pesticide» is also technically defined as the formulated product, I suggest you use «Pesticide» in stead of «active substance» and «active substance in PPP», since this use of the term is so common. You may clarify this use of the phrase pesticide in the introduction. This will also harmonize with your Title, where you use «Pesticide». (Piperonyl butoxide, being a synergist and not a pesticide, may need some remarks). If you decide to keep «active substances» as a term, the use of «PPP» in the text need to be critically evaluated to see if it is used as per definition. Has been changed

-Line 20, 93, 227, 261, 502: How many active substances were detected in the samples. Different things are stated (20, 21 or 23 substances). Has been corrected in the manuscript

-Please check the wording of every Figure text/Table text (headings), there are many non-precise headings. E.g. Table A1: «List French wines tested», which should be «A list of the French wines tested». OK

-The paper puts weight on findings above LOD and findings above LOQ, hence it is necessary to know the respective LOD and LOQs and this needs to be clarified better (for GC-ECD analysis).

Here a correction has been given and the GC-ECD analysis has been taken out of the manuscript. This due to the insufficent possitive detections with this method.

-The language and clarity in the sentences can be improved. See separate pages at the end of this document, with some of my suggested amendments in blue text. OK

Minor aspects in Introduction:

-Line 78: please remove «celery leaves, and majoram» OK

-Line 83-84: it is unclear what is meant with this sentence. Please insert an update. OK

Minor and major aspects in Material & Methods:

-Please update in Figure 1: 

- The Legend text «The distribution of the 84 wine bottles by year of production and region of production in France». OK

- The dark blue and the blue columns are quite similar in colour. Can the dark blue colur be edited to another colour? OK

-Line 120-121: please remove «The cartridge effectively retained all PPPs present in the wine», as this is not known – the wine could contain other pesticides not analyzed for. OK

-Line 121-122: was the cartridge rinsed with hexane and then eluted with AcN? It seems later in the text that the cartridge was eluted with 2 ml hexane and 8 ml acetonitrile. Hence the hexane is not a rinse, but use as an eluent? Please insert a clarification. Yes the hexane is an eluent. Text added in the manuscript.

-Line 129, 132 a.o.: Please write «wine sediment» in stead of only «sediment», as many readers will associate «sediment» with sediments in streams and rivers. This relates to several places in the text. Please insert updates. OK

Line 132-141. When was acetonitrile added to the wine sediment sample? Please insert. Corrected in the manuscript

Line 142: How was the evaporation performed. Please insert. Evaporation was perferomed with Rotavapor machine. 

Line 157: Was matrix-matched calibration performed, or were standards solved in solvent only? Please insert description. Yes, a recovery correction factor was applied to the results to account for the matrix effects, which were significant in this case. Despite these effects, the method demonstrated adequate reproducibility. To evaluate the matrix, organic wine and tested on Carbendazim, providing a representative sample for the analytical performance. A multi-residue method was employed to address the challenge that one method cannot extract all pesticides, with different polarity/solubility, in a single solvent, at 100% efficiency. This approach ensures comprehensive detection and quantification of various pesticides, compensating for the limitations inherent in solvent-based extraction methods.

2.2 LC-MS/MS Analysis: 

Line 166-167: triple quadrupole mass spectrometry (not double mass). I suppose triple quadrupole was used? Please specify the model of the LC-MS/MS. Model: Xevo TQD Waters

Line 177: The LOQs for the LC-MS/MS method is stated below Table A.2, but the LOQs for the GC-ECD method is not stated enywhere. Please add. Please state LOQs in µg/L (not mg/L). 

Here a correction has been given and the GC-ECD analysis has been taken out of the manuscript. This due to the insufficent possitive detections with this method.

Line 182-183: The decription of how the quantification was performed is very unclear. Please rewrite. Was processed (worked-up) matrix calibration standards used for the quantification? Or were some pesticides quantified with solvent standards/matrix-matched standards and some pesticides with processed matrix calibration standards? Please specify in the text. May add the range of the calibration standards. Text in the manuscript has been rewritten. 

How many pesticides were targeted in the LC-MS/MS method? Please specify in the text. Text in the manuscript was adapted.

Which of the detected pesticides in the wine were detected with the LC-MS/MS method. Please specify in the text. All 21 pesticedes detected were coming from this method. They are listed in Figure 2.

-What was the LO – LO means ‘Limit of ... ‘ 

-Please add a list of all the pesticides in the analytical methods, either in 2.2 LC-MS/MS section (and 2.3 GC-ECD section) or in an Appendix.

The list is added in the appendix table A12

2.3 GC-ECD Analysis:

Here a correction has been given and the GC-ECD analysis has been taken out of the manuscript. This due to the insufficent possitive detections with this method.

Please specify the model of the GC-ECD.

Please add a description on how the quantification with calibration standards was performed. May add the range of the calibration standards.

Which of the detected pesticides in the wine were detected with the GC-ECD method. Please specify in the text. 

3 Results:

-Line 230: Explain why cadusafos was non-quantifiable (due to cadusafos levels being below the LOQ). Has been corrected

-Line 249: It should be commented that carbendazim is also a metabolite of thiophanate-methyl and benomyl. Have any of these been approved in viticulture in France?

This is not really the research topic for this analysis. This research was not aimed to address whether the pesticides in question were permitted in France. Instead, the primary objective was to highlight the need for re-evaluating collector's food products, such as vintage wines, against current, stringent food safety standards. Additionally, the aim was to increase public awareness about the potential pesticide residues in these older products. Contemporary food regulations do not permit residues of substances that are banned today, even if those substances were previously allowed. This study seeks to illuminate the issue of historical contamination and offers a new perspective on assessing wines compared to other food items. This statements have been incorporated in the conclusions.

Line 297: This is a valid point (on systemic vs non-systemic/contact). However, you may add some comments on application rates and application dosages, which may (usually) be higher for fungicides than for insecticides? The fields being treated more often with fungicides than with insecticides, and at higher dosages for the fungicides? Text has been added to the manuscript

-Line 307: Explain the colours used in Table A.6. What/why has some values been considered «high levels» whereas other leves have been considered low? (e.g. 

The table has been revised and edited to underline if the values of 0,01 mg/L and 5mg/L and the contents values accroding to EU rules have been or not have been exceeded. Legend explaining color coding has been added to the tables A.6 and A.7. 

-Line 333 (Figure 6), line 336, Figure 10 a.o.: It is better to use the phrase «production year of the wine» in stead of «the year of wine», «over the years». «Age of the wine» can be used. Adapted

Line 344: The analysis method(s) did not include analysis of obsolete and banned pesticides such as DDT and metabolites, aldrin and other POPs. Hence residues of these in the (old) wine may have been missed. Please insert a comment on this in the paper. You may also want to comment on metabolites not being included in the analysis method. Comment has been added to the manucript.

Line 398: A comment: the lower pesticide levels in the wine sediments as compared to the wine liquid phase – even though many of the fungicides exhibit high sorption coefficients - may also be due to the ethanol content in the wine, as the ethanol increases the pesticides’ solubility in the wine liquid. Text has been added to the manuscript. 

-Line 403: rewrite «which was not possible in wine». It is an unclear statement. Has been changed

-Line 423: The Table A.11 refers to the pesticides detected in one of the analyzed wine bottles, not all botles. Please correct. Has been changed

Line 445: «Pesticide degradation adheres to a first-order kinetic model..». Pesticide degradation may also be bi-phasic, e.g. fast degradation of dissolved pesticide with a slow degradation of sorbed pesticide. It depends on the system/matrix. Please modify. (e.g. Pesticide degradation often adheres to a first-order kinetic model..». Text has been added to the manuscript.

Line 446: «This model implies that as pesticide levels diminish, the time required for further degradation extends» What do you mean here? In a single-first-order linear degradation (SFO), the degradation rate is constant throughout. Please remove, or rephrase. Text has been added to the manuscript.

-Line 449: this is a good point.

-Line 484: «Most pesticides are banned in grape production,..». Rather rephrase this sentence to e.g. «Pesticides not approved for use in grapes have default MRL values set at 0.01 mg/kg or 0.05 mg/kg.» Has been changed

Line 485: MRLs in grapes are calculated into MRL in wine, using a processing factor, to account for pesticide residues in the final product (wine). There exists calculated EU MRLs in wine. Correct. We are mentioning this database (EU database) in references – reference number 36. 

Line 488: «Considering the quantity and variety of pesticides detected in the older wines, these products fail to meet current European regulatory standards and should normally be taken out of the market». This needs more explanation and justification. You are not discussing the MRL levels with the levels detected in the individual wine samples here? (i.e. Table A6). Which samples exceeded the MRL levels? «The quantity of pesticides» means the number of pesticides detected, not their pesticide concentrations. The number/sum of pesticides are not regulated? 

Actually, the pesticide levels in the wine samples are not really discussed in the paper. Table A.6 is presented but the levels are not really discussed in the Results and Discussion section? If you choose not to discuss the levels found compared to the calculated MRLs in wine, it is OK. But then you cannot make the statement of Considering the quantity and variety of pesticides detected in the older wines, these products fail to meet current European regulatory standards and should normally be taken out of the market». 

Agreed. Statement in the text has been adapted and part of disscussion of the result has been added about MRL levels.

Line 521-530: These are valid points. Though I lack some coherence (or better explanations) between the findings and the conclusions.

Text in the manuscript has been rewritten

Table A.4:

-Please state the mean pesticide concentration of positive samples as µg/L (not mg/L). (Since in Table 1.6. concentrations are stated in µg/L). Are the concentrations in table A.4 the sum of residues found in wine sediment and in wine liquid? Table A.6 was adapted to mg/L as to match the table A.4. As official pesticide data are usllay set to mg/L. The values in table A.4 are mean not the sum. 

-Please remove the pesticides that were not-quantifiable; the Table heading says the Table contains the pesticides in wine with concentrations higher than LOQ. Table A4 was adapted

-Make sure the mean concentrations are only for the positive samples (as stated), and that the mean does not include the zero samples? I note that many of the mean concentrations are below the LOQ of 1 µg/L for LC-MS/MS compounds? Has been revised

---

## [Editor Report · Decision Letter 2]

3 Dec 2024

PONE-D-24-17146R2The temporal variation in pesticide concentrations within matured French wines.PLOS ONE

Dear Dr. Spanoghe,

Thank you for submitting your manuscript to PLOS ONE. After careful consideration, we feel that it has merit but does not fully meet PLOS ONE’s publication criteria as it currently stands. Therefore, we invite you to submit a revised version of the manuscript that addresses the points raised during the review process. **Please submit the response to the reviewers more diligently. There are four manuscript files in the PDF. Please include only the relevant files (one in track change mode and the other without track change mode). Additionally, please clarify the GC-ECD data.** 

We look forward to receiving your revised manuscript.

Kind regards,

Totan Adak

Academic Editor

PLOS ONE

**Journal Requirements:**

**Additional Editor Comments:**

Please submit the response to the reviewers more diligently. There are four manuscript files in the PDF. Please include only the relevant files (one in track change mode and the other without track change mode). Additionally, please clarify the GC-ECD data.

---

## [Author Response · Author response to Decision Letter 2]

12 Dec 2024

Dear reviewer, we thank you for reading our manuscript carefully and for the comments that shed a new light on some of the inconsitencies of the text. Please find our answers below and a manucript version with track changes.

Manuscript Number PONE-D-24-17146R1

The temporal variation in pesticide concentrations within matured French wines.

General comments from reviewer:

-It is a g good idea to screen bottled wine for pesticide residues. The range of pesticides analyzed is fair, which adds to the conclusions that can be drawn (although I don’t have a very good overview of pesticides approved in grapes), but the pesticide list need to be clarified better (e.g. in Appendix). Has been added to the appendix

-It can be confusing when you refer to «active substance» in figures, but use «PPP» in the text. The «PPP» should in most cases be understood as «active substance in PPP». A PPP is the formulated product, that contains one or more active substances and co-formulants. You refer to [12] Sykalia et al (2024), and they use the term «Pesticide» for all the active substances in their paper – as is also commonly used in other papers. Although «pesticide» is also technically defined as the formulated product, I suggest you use «Pesticide» in stead of «active substance» and «active substance in PPP», since this use of the term is so common. You may clarify this use of the phrase pesticide in the introduction. This will also harmonize with your Title, where you use «Pesticide». (Piperonyl butoxide, being a synergist and not a pesticide, may need some remarks). If you decide to keep «active substances» as a term, the use of «PPP» in the text need to be critically evaluated to see if it is used as per definition. Has been changed

-Line 20, 93, 227, 261, 502: How many active substances were detected in the samples. Different things are stated (20, 21 or 23 substances). Has been corrected in the manuscript

-Please check the wording of every Figure text/Table text (headings), there are many non-precise headings. E.g. Table A1: «List French wines tested», which should be «A list of the French wines tested». OK

-The paper puts weight on findings above LOD and findings above LOQ, hence it is necessary to know the respective LOD and LOQs and this needs to be clarified better (for GC-ECD analysis).

Here a correction has been given and the GC-ECD analysis has been taken out of the manuscript. This due to the insufficent possitive detections with this method.

-The language and clarity in the sentences can be improved. See separate pages at the end of this document, with some of my suggested amendments in blue text. OK

Minor aspects in Introduction:

-Line 78: please remove «celery leaves, and majoram» OK

-Line 83-84: it is unclear what is meant with this sentence. Please insert an update. OK

Minor and major aspects in Material & Methods:

-Please update in Figure 1:

- The Legend text «The distribution of the 84 wine bottles by year of production and region of production in France». OK

- The dark blue and the blue columns are quite similar in colour. Can the dark blue colur be edited to another colour? OK

-Line 120-121: please remove «The cartridge effectively retained all PPPs present in the wine», as this is not known – the wine could contain other pesticides not analyzed for. OK

-Line 121-122: was the cartridge rinsed with hexane and then eluted with AcN? It seems later in the text that the cartridge was eluted with 2 ml hexane and 8 ml acetonitrile. Hence the hexane is not a rinse, but use as an eluent? Please insert a clarification. Yes the hexane is an eluent. Text added in the manuscript.

-Line 129, 132 a.o.: Please write «wine sediment» in stead of only «sediment», as many readers will associate «sediment» with sediments in streams and rivers. This relates to several places in the text. Please insert updates. OK

Line 132-141. When was acetonitrile added to the wine sediment sample? Please insert. Corrected in the manuscript

Line 142: How was the evaporation performed. Please insert. Evaporation was perferomed with Rotavapor machine.

Line 157: Was matrix-matched calibration performed, or were standards solved in solvent only? Please insert description. Yes, a recovery correction factor was applied to the results to account for the matrix effects, which were significant in this case. Despite these effects, the method demonstrated adequate reproducibility. To evaluate the matrix, organic wine and tested on Carbendazim, providing a representative sample for the analytical performance. A multi-residue method was employed to address the challenge that one method cannot extract all pesticides, with different polarity/solubility, in a single solvent, at 100% efficiency. This approach ensures comprehensive detection and quantification of various pesticides, compensating for the limitations inherent in solvent-based extraction methods.

2.2 LC-MS/MS Analysis:

Line 166-167: triple quadrupole mass spectrometry (not double mass). I suppose triple quadrupole was used? Please specify the model of the LC-MS/MS. Model: Xevo TQD Waters

Line 177: The LOQs for the LC-MS/MS method is stated below Table A.2, but the LOQs for the GC-ECD method is not stated enywhere. Please add. Please state LOQs in µg/L (not mg/L).

Here a correction has been given and the GC-ECD analysis has been taken out of the manuscript. This due to the insufficent possitive detections with this method.

Line 182-183: The decription of how the quantification was performed is very unclear. Please rewrite. Was processed (worked-up) matrix calibration standards used for the quantification? Or were some pesticides quantified with solvent standards/matrix-matched standards and some pesticides with processed matrix calibration standards? Please specify in the text. May add the range of the calibration standards. Text in the manuscript has been rewritten.

How many pesticides were targeted in the LC-MS/MS method? Please specify in the text. Text in the manuscript was adapted.

Which of the detected pesticides in the wine were detected with the LC-MS/MS method. Please specify in the text. All 21 pesticedes detected were coming from this method. They are listed in Figure 2.

-What was the LO – LO means ‘Limit of ... ‘

-Please add a list of all the pesticides in the analytical methods, either in 2.2 LC-MS/MS section (and 2.3 GC-ECD section) or in an Appendix.

The list is added in the appendix table A12

2.3 GC-ECD Analysis:

Here a correction has been given and the GC-ECD analysis has been taken out of the manuscript. This due to the insufficent possitive detections with this method.

Please specify the model of the GC-ECD.

Please add a description on how the quantification with calibration standards was performed. May add the range of the calibration standards.

Which of the detected pesticides in the wine were detected with the GC-ECD method. Please specify in the text.

3 Results:

-Line 230: Explain why cadusafos was non-quantifiable (due to cadusafos levels being below the LOQ). Has been corrected

-Line 249: It should be commented that carbendazim is also a metabolite of thiophanate-methyl and benomyl. Have any of these been approved in viticulture in France?

This is not really the research topic for this analysis. This research was not aimed to address whether the pesticides in question were permitted in France. Instead, the primary objective was to highlight the need for re-evaluating collector's food products, such as vintage wines, against current, stringent food safety standards. Additionally, the aim was to increase public awareness about the potential pesticide residues in these older products. Contemporary food regulations do not permit residues of substances that are banned today, even if those substances were previously allowed. This study seeks to illuminate the issue of historical contamination and offers a new perspective on assessing wines compared to other food items. This statements have been incorporated in the conclusions.

Line 297: This is a valid point (on systemic vs non-systemic/contact). However, you may add some comments on application rates and application dosages, which may (usually) be higher for fungicides than for insecticides? The fields being treated more often with fungicides than with insecticides, and at higher dosages for the fungicides? Text has been added to the manuscript

-Line 307: Explain the colours used in Table A.6. What/why has some values been considered «high levels» whereas other leves have been considered low? (e.g.

The table has been revised and edited to underline if the values of 0,01 mg/L and 5mg/L and the contents values accroding to EU rules have been or not have been exceeded. Legend explaining color coding has been added to the tables A.6 and A.7.

-Line 333 (Figure 6), line 336, Figure 10 a.o.: It is better to use the phrase «production year of the wine» in stead of «the year of wine», «over the years». «Age of the wine» can be used. Adapted

Line 344: The analysis method(s) did not include analysis of obsolete and banned pesticides such as DDT and metabolites, aldrin and other POPs. Hence residues of these in the (old) wine may have been missed. Please insert a comment on this in the paper. You may also want to comment on metabolites not being included in the analysis method. Comment has been added to the manucript.

Line 398: A comment: the lower pesticide levels in the wine sediments as compared to the wine liquid phase – even though many of the fungicides exhibit high sorption coefficients - may also be due to the ethanol content in the wine, as the ethanol increases the pesticides’ solubility in the wine liquid. Text has been added to the manuscript.

-Line 403: rewrite «which was not possible in wine». It is an unclear statement. Has been changed

-Line 423: The Table A.11 refers to the pesticides detected in one of the analyzed wine bottles, not all botles. Please correct. Has been changed

Line 445: «Pesticide degradation adheres to a first-order kinetic model..». Pesticide degradation may also be bi-phasic, e.g. fast degradation of dissolved pesticide with a slow degradation of sorbed pesticide. It depends on the system/matrix. Please modify. (e.g. Pesticide degradation often adheres to a first-order kinetic model..». Text has been added to the manuscript.

Line 446: «This model implies that as pesticide levels diminish, the time required for further degradation extends» What do you mean here? In a single-first-order linear degradation (SFO), the degradation rate is constant throughout. Please remove, or rephrase. Text has been added to the manuscript.

-Line 449: this is a good point.

-Line 484: «Most pesticides are banned in grape production,..». Rather rephrase this sentence to e.g. «Pesticides not approved for use in grapes have default MRL values set at 0.01 mg/kg or 0.05 mg/kg.» Has been changed

Line 485: MRLs in grapes are calculated into MRL in wine, using a processing factor, to account for pesticide residues in the final product (wine). There exists calculated EU MRLs in wine. Correct. We are mentioning this database (EU database) in references – reference number 36.

Line 488: «Considering the quantity and variety of pesticides detected in the older wines, these products fail to meet current European regulatory standards and should normally be taken out of the market». This needs more explanation and justification. You are not discussing the MRL levels with the levels detected in the individual wine samples here? (i.e. Table A6). Which samples exceeded the MRL levels? «The quantity of pesticides» means the number of pesticides detected, not their pesticide concentrations. The number/sum of pesticides are not regulated?

Actually, the pesticide levels in the wine samples are not really discussed in the paper. Table A.6 is presented but the levels are not really discussed in the Results and Discussion section? If you choose not to discuss the levels found compared to the calculated MRLs in wine, it is OK. But then you cannot make the statement of Considering the quantity and variety of pesticides detected in the older wines, these products fail to meet current European regulatory standards and should normally be taken out of the market».

Agreed. Statement in the text has been adapted and part of disscussion of the result has been added about MRL levels.

Line 521-530: These are valid points. Though I lack some coherence (or better explanations) between the findings and the conclusions.

Text in the manuscript has been rewritten

Table A.4:

-Please state the mean pesticide concentration of positive samples as µg/L (not mg/L). (Since in Table 1.6. concentrations are stated in µg/L). Are the concentrations in table A.4 the sum of residues found in wine sediment and in wine liquid? Table A.6 was adapted to mg/L as to match the table A.4. As official pesticide data are usllay set to mg/L. The values in table A.4 are mean not the sum.

-Please remove the pesticides that were not-quantifiable; the Table heading says the Table contains the pesticides in wine with concentrations higher than LOQ. Table A4 was adapted

-Make sure the mean concentrations are only for the positive samples (as stated), and that the mean does not include the zero samples? I note that many of the mean concentrations are below the LOQ of 1 µg/L for LC-MS/MS compounds? Has been revised

---

## [Editor Report · Decision Letter 3]

22 Dec 2024

The temporal variation in pesticide concentrations within matured French wines.

PONE-D-24-17146R3

Dear Dr. Spanoghe,

We’re pleased to inform you that your manuscript has been judged scientifically suitable for publication and will be formally accepted for publication once it meets all outstanding technical requirements.

Kind regards,

Totan Adak

Academic Editor

PLOS ONE
---

## [Editor Report · Acceptance letter]

24 Jan 2025

PONE-D-24-17146R3 

PLOS ONE

Dear Dr. Spanoghe, 

I'm pleased to inform you that your manuscript has been deemed suitable for publication in PLOS ONE. Congratulations! Your manuscript is now being handed over to our production team.

Kind regards, 

on behalf of

Dr. Totan Adak 

Academic Editor

PLOS ONE